# A Cartesian-$3j$ Framework for Machine Learning Interatomic Potentials

**Zemin Xu** [1 2]  **Chenyu Wu** [1 2]  **Wenbo Xie** [1]  **P. Hu** [1]

## Abstract

Machine learning interatomic potentials (MLIPs) have brought substantial gains in the extrapolation capability in computational chemistry. However, most equivariant models are typically built with spherical tensors (STs), while Cartesian tensor formulations remain less developed despite their natural alignment with atomic coordinates and tensorial targets. In this work, we develop a Cartesian framework for irreducible Cartesian tensors (ICTs) by introducing the `Cartesian-3j` symbol and Cartesian Generalized Clebsch-Gordan Coefficients, which serve as direct analogues of the `Wigner-3j` symbol and Generalized Clebsch-Gordan coefficients defined for ST coupling. We extend the `e3nn` library to support ICT product, and use this framework to build Cartesian counterparts of `MACE`, `NequIP`, and `Allegro`, allowing the first controlled comparison where architectures are held fixed and only the tensor basis is changed. Our experiments show that irreducible Cartesian models can achieve accuracy comparable to spherical counterparts, but direct Cartesianization incurs unfavorable compute and memory scaling, motivating dedicated Cartesian architectural choices. Leveraging ICTs and our framework, we introduce `TACE-v1-OAM-M` and demonstrate that it achieves competitive performance on Matbench Discovery compared to state-of-the-art ST models.

## 1. Introduction

Machine learning interatomic potentials have become a core tool for accelerating atomistic simulation across chemistry and materials science. Based on data labeled from first-principles calculations, atomic environments are encoded as learned tensors, and the desired physical properties are obtained through an appropriate readout mechanism. Modern MLIPs are no longer limited to predicting energies, forces, stresses, and virials. Increasingly, they are used to model a broad spectrum of physical quantities, including dipole moments, polarization, polarizability, Born effective charges, nuclear shielding tensors, elastic tensors, dielectric tensors, piezoelectric tensors and related response properties (Zhang et al., 2020; Gastegger et al., 2021; Xu et al., 2024; Wen et al., 2024; Falletta et al., 2025; Martin et al., 2025; Xu et al., 2026). These expanded capabilities greatly accelerate materials discovery, reaction mechanism studies, field-dependent dynamical behavior, spectral simulations, and related applications.

Most equivariant MLIPs directly operate on irreducible representations of the O(3) group, i.e., they construct models using STs. Models such as NequIP, Allegro, and MACE (Batzner et al., 2022; Batatia et al., 2022; Musaelian et al., 2023) introduced physically informed inductive biases that markedly improved generalization and data efficiency. Despite this success, two practical issues motivate alternative model architecture. First, the computational burden associated with standard Clebsch-Gordan Tensor Products/Spherical Tensor Products (CGTP/STP) highlights the need for more effective strategies that balance accuracy and computational cost. Second, atomistic inputs (Cartesian coordinates) and physical properties (Cartesian tensors) are naturally represented in Cartesian space, motivating the question of whether ICTs can provide a cleaner or more efficient path to equivariant modeling.

Recent works have explored formulations in Cartesian space. However, because the irreducible components of Cartesian tensors (CTs) are inherently intertwined, reducible Cartesian tensor products (RCTPs) and contractions (RCTCs), while simpler and more intuitive to use, cannot separately embed learnable weights for different irreducible components, thereby leading to a loss of accuracy. In particular, the lack of direct analogues of the Wigner-$3j$ and their higher-order generalizations in Cartesian space prevents a direct replacement for tensor couplings, making controlled comparisons between spherical and Cartesian models difficult when model architectures are fixed. Moreover, the computational cost of CTs increases exponentially with tensor rank when compared with STs, which raises significant chal-

[1]School of Physical Science and Technology, ShanghaiTech University, Shanghai, China [2]School of Chemistry, Nanjing University, Nanjing, China. Correspondence to: Wenbo Xie <xiewb1@shanghaitech.edu.cn>.

*Proceedings of the $43^{rd}$ International Conference on Machine Learning*, Seoul, South Korea. PMLR 306, 2026. Copyright 2026 by the author(s).

lenges for the design of Cartesian models and requires more sophisticated strategies to maintain efficiency.

**Contributions.** Our main contributions are as follows. (i) We establish the theoretical foundations of Cartesian equivariant models by introducing Cartesian-$3j$ symbol and Cartesian Generalized Clebsch-Gordan Coefficients (Batatia et al., 2022). (ii) Based on this framework, we extend the e3nn package (Thomas et al., 2018; Weiler et al., 2018; Kondor et al., 2018) to support irreducible Cartesian tensor product (ICTP) and provide a principled implementation of irreducible Cartesian tensor contraction (ICTC). (iii) Leveraging the extended e3nn (cartnn), we construct Cartesian versions of NequIP, Allegro, and MACE based on ICTP, enabling the first controlled comparison in which the network architecture and hyperparameters are fixed and only the tensor basis (ST vs. ICT) is varied. Through these controlled experiments, we show that, under matched architectures, ICT-based models achieve accuracy comparable to their spherical counterparts. However, naive "Cartesianized" versions suffer from unfavorable compute and memory scaling, highlighting the need for dedicated Cartesian design choices. (iv) To the best of our knowledge, we establish the current best-practice paradigm for Cartesian equivariant models and train a series of universal machine learning interatomic potentials (uMLIPs), which are benchmarked on Matbench Discovery (Riebesell et al., 2025). The results demonstrate the potential of Cartesian models. Although they do not yet achieve state-of-the-art (SOTA) performance due to the absence of efficient kernel fusion support, they still exhibit competitive performance relative to spherical models.

## 2. Related Work

**Reducible Methods.** HotPP (Wang et al., 2024) introduced arbitrary rank RCTP and RCTC, which enabled the model to surpass the speed of spherical architectures such as NequIP (Batzner et al., 2022) under low rank settings. CACE (Cheng, 2024) extended the invariant framework of REANN (Zhang et al., 2021) and provided a systematic approach for representing higher body interactions in Cartesian space. CAMP (Wen et al., 2025) further exploited the symmetries of RCTP and RCTC, which allowed the model to restrict the admissible paths and ensured complete symmetry. Although these networks have achieved meaningful progress in Cartesian formulations, each of them remains limited in comparison with spherical models in at least one important dimension. In principle, high rank Cartesian constructions are not expected to outperform spherical approaches, yet even within the regime of low rank representations, the Cartesian models developed thus far have not truly surpassed their spherical counterparts.

**Irreducible Methods.** TensorNet (Simeon & De Fabritiis, 2023) represents an elegant architectural design, as it employs ICTs of at most rank 2. Working directly with irreducible components of rank-2 and below is theoretically straightforward, and this choice also offers clear advantages in both computational speed and memory usage. To the best of our knowledge, only three models are currently capable of manipulating irreducible Cartesian components at arbitrary rank. These models are TACE (Xu et al., 2026), ICT potential (Zaverkin et al., 2024), and CarNet (Chen et al., 2026). TACE is built upon the irreducible Cartesian tensor decomposition theory (Shao et al., 2025), while ICT potential and CarNet rely on the irreducible Cartesian tensor product framework (Lehman & Parke, 1989). From our perspective, the decomposition-based formulation is more elegant because it allows a Cartesian tensor to be manipulated and decomposed in a flexible manner and provides the ability to convert freely between Cartesian and spherical bases. The latter tensor product theory is effective, yet it still exhibits limitations in lower weight and often requires separate analytic expressions for each specific tensor product in practice. This creates additional complications for implementation. For example, ICT potential implements irreducible Cartesian operations only up to rank-3, whereas TACE is capable of handling arbitrary rank. Importantly, the results reported for the ICT potential (Zaverkin et al., 2024) show that Cartesian tensor operations can offer a computational advantage for tensor ranks up to 4.

**Foundational models.** After evaluating the strengths and weaknesses of the Cartesian model, we proposed design principles for the Cartesian model and trained a series of uMLIPs. We then compare our approach with representative uMLIPs across key metrics in Matbench Discovery, including the F1 score related to formation energy and thermal conductivity $\kappa$, which depends on higher-order derivatives of the potential energy surface. (Riebesell et al., 2025; Fu et al., 2025; Lee et al., 2025; Tan et al., 2025; Lysogorskiy et al., 2026; Rhodes et al., 2025; Park et al., 2024; Zhang et al., 2025; Batatia et al., 2025b;c; Yang et al., 2024).

## 3. Background

As part of the Cartesian framework, we find it necessary to briefly review some key concepts. For a more detailed introduction, we refer the reader to (Lehman & Parke, 1989; Zaverkin et al., 2024; Shao et al., 2025; Xu et al., 2026).

### 3.1. Cartesian tensor

A generic Cartesian tensor of rank-$\nu$, denoted as ${}^{\nu}\mathbf{T}$, is a multilinear object with $3^{\nu}$ components. The term "generic Cartesian tensor" refers to one with no imposed symmetry or traceless constraints among its indices. In general, such

tensors are reducible under O(3), and each irreducible component is associated with a definite weight $\ell$ (note that this "weight" is not the "learning weight" in machine learning). A generic Cartesian tensor $^{\nu}\mathbf{T}$ can be expressed as a sum of ICTs of the same rank but different weight:

$$^{\nu}\mathbf{T} = \sum_{\ell,q} {}^{(\nu;\ell;q)}\mathbf{T}, \tag{1}$$

where each $^{(\nu;\ell;q)}\mathbf{T}$ is an irreducible component with $2\ell + 1$ degrees of freedom, of rank $\nu$ and weight $\ell$, and $q$ denotes the multiplicity for given values of $\nu$ and $\ell$.

### 3.2. Irreducible Cartesian tensor decomposition

It is well known that a generic rank-2 Cartesian tensor can be decomposed into ICTs with weights $\ell = 2$, $\ell = 1$, and $\ell = 0$. Indeed, ICTs can be obtained using irreducible Cartesian tensor decomposition (ICTD) matrices, which were previously available only for ranks up to $\nu = 5$ (Coope et al., 1965; Andrews & Ghoul, 1982; Bonvicini, 2024), and whose construction has traditionally been computationally expensive. Recently, however, analytical orthogonal ICTD matrices have been derived that enable efficient construction at arbitrary rank (Shao et al., 2025). We denote the corresponding ICTD operator and its associated decomposition matrix by $\mathcal{T}$. The overall decomposition can be expressed as follows, where $\text{vec}(\cdot)$ denotes the vectorization (flattening) of a Cartesian tensor:

$$\text{vec}\left( {}^{(\nu;\ell;q)}\mathbf{T} \right) = {}^{(\nu;\ell;q)}\mathcal{T} \, \text{vec}\left( {}^{\nu}\mathbf{T} \right). \tag{2}$$

The inspiration for ICTD comes from the fact that the Cartesian tensor product space and the spherical direct-sum spaces differ only by a change of basis. The most important feature of ICTD operator is that $\sum_{l,q} {}^{(\nu;\ell;q)}\mathcal{T} = I$, where $I$ denotes the identity matrix. In addition, it also exhibits orthogonality, a matrix rank of $2\ell + 1$, O(3)-equivariance, and other related properties.

### 3.3. Cartesian harmonics

Given the normalized vector $\hat{\mathbf{r}}_{ij} = \mathbf{r}_{ij}/r_{ij}$ (from source atom $j$ to target atom $i$), Cartesian harmonics are defined as follows:

$$^{(\nu;\nu;1)}\mathbf{E}_{ij} = \frac{(2\nu - 1)!!}{\nu!}$$
$$\times {}^{(\nu;\nu;1)}\mathcal{T}\left( \underbrace{\hat{\mathbf{r}}_{ij} \otimes \cdots \otimes \hat{\mathbf{r}}_{ij}}_{\nu \text{ times}} \right). \tag{3}$$

It is important to note that Cartesian harmonics are symmetric traceless tensors, i.e., irreducible Cartesian tensors. In general, directly taking outer products may leave residual trace components. Cartesian harmonics can be obtained

by recursively removing traces, as implemented in our cartnn (Toth & Turyshev, 2022), via an alternative formulation proposed in (Applequist, 1989), or through the ICT decomposition approach (Xu et al., 2026). The constant factor here ensures that $^{(\nu;\nu;1)}\mathbf{E}_{ij}$ contracts appropriately to $\hat{\mathbf{r}}_{ij}$.

## 4. Cartesian framework

### 4.1. Cartesian-$3j$

We denote the path matrix (Shao et al., 2025) by $^{(\nu;\ell;q)}C$. Its role is to serve as the transformation matrix between Cartesian tensors and spherical tensors. Correspondingly, its inverse transformation can be constructed through the transpose matrix $^{(\nu;\ell;q)}C^{T}$. In particular, for the special case with $\text{rank} = \text{weight}$, we abbreviate the path matrix as $^{(\ell)}C$. Its matrix elements are denoted by $^{(\ell)}C_{mM}$, where the Cartesian index $m$ runs over the $3^{\ell}$ Cartesian tensor components, while the spherical index $M$ runs over the $2\ell + 1$ spherical tensor components. The path matrix is generated according to the parentage scheme through chain-like contractions with CG coefficients followed by normalization, and the ICTD matrix is defined as:

$$^{(\nu;\ell;q)}\mathcal{T} = {}^{(\nu;\ell;q)}C \, {}^{(\nu;\ell;q)}C^{T}. \tag{4}$$

We provide two interpretations of the irreducible Cartesian tensor product; the second interpretation and its corresponding results are presented in Appendix A. Consider two STs, $^{(\ell_1)}\text{ST}$ and $^{(\ell_2)}\text{ST}$. These STs can be viewed as being obtained from the Cartesian tensors $^{(\ell_1;\ell_1;1)}\text{T}$ and $^{(\ell_2;\ell_2;1)}\text{T}$ through path-matrix transformations. We can also apply an additional path-matrix transformation to the output $^{(\ell_3)}\text{ST}$. In practical implementations, the computation can be simplified by first contracting the Wigner-$3j$ symbols with the three path matrices, thereby yielding the Cartesian-$3j$ symbols directly:

$$Z^{\ell_3 m_3}_{\ell_1 m_1, \ell_2 m_2} = \sum_{M_1, M_2, M_3} {}^{(\ell_3)}C_{m_3 M_3} \begin{pmatrix} L_1 & L_2 & L_3 \\ M_1 & M_2 & M_3 \end{pmatrix}$$
$$\times {}^{(\ell_1)}C_{m_1 M_1} \, {}^{(\ell_2)}C_{m_2 M_2}, \tag{5}$$

where $\begin{pmatrix} L_1 & L_2 & L_3 \\ M_1 & M_2 & M_3 \end{pmatrix}$ denotes the Wigner-$3j$ symbol. In practical applications, we usually apply L2 norm normalization to maintain stable variance.

### 4.2. Cartesian Generalized Clebsch-Gordan Coefficients

It is known that the generalized CG coefficients proposed in MACE (Batatia et al., 2022) significantly reduce the computational cost of channel-wise tensor products between identical tensors in spherical space. By precomputing the generalized CG coefficients and sequentially contracting

them with node features and element-dependent trainable weights, MACE leverages the advantages of the ACE framework to reduce the number of message-passing layers to two while greatly lowering both computation time and memory consumption. However, no analogue of the Wigner-$3j$ symbols previously existed in Cartesian space. For this reason, ICT potential (Zaverkin et al., 2024) also concluded that certain computations can't be precomputed in Cartesian space. Consequently, in both ICT potential and TACE (Zaverkin et al., 2024; Xu et al., 2026), the product basis is computed from scratch during the forward pass. Nonetheless, if the Wigner-$3j$ symbols are replaced by their Cartesian counterparts, the Cartesian Generalized CG Coefficients can be obtained directly in Cartesian space with ease:

$$\mathcal{Z}_{l_1 m_1, .., l_n m_n}^{LM} = Z_{l_1 m_1, l_2 m_2}^{L_2 M_2} Z_{L_2 M_2, l_3 m_3}^{L_3 M_3} \\ \cdots Z_{L_{N-1} M_{N-1}, l_N m_N}^{L_N M_N},$$ (6)

where $L \equiv (L_2, .., L_N)$, $|l_1 - l_2| \le L_2 \le l_1 + l_2 \ \forall \ i \ge 3 |L_{i-1} - l_i| \le L_i \le L_{i-1} + l_i$, and $M_i \in \{m_i| - l_i \le m_i \le l_i\}$.

### 4.3. Tensor Product and Contraction

Let $^\ell\mathbf{T}$ denote a rank-$\ell$ Cartesian tensor with components $\mathbf{T}_{a_1 \ldots a_\ell}$, where each index $a$ runs over the Cartesian dimensions $x, y, z$. Here, the symbol $a$ is a dummy index and may be replaced by any other valid index symbol. The RCTP corresponds to the case where no indices are contracted between two tensors. Given two tensors $^{\ell_1}\mathbf{T}$ and $^{\ell_2}\mathbf{T}$, their RCTP is defined as

$$^{\ell_1+\ell_2}\mathbf{T}_{a_1 \ldots a_{\ell_1} \ b_1 \ldots b_{\ell_2}} = {}^{\ell_1}\mathbf{T}_{a_1 \ldots a_{\ell_1}} \cdot {}^{\ell_2}\mathbf{T}_{b_1 \ldots b_{\ell_2}}.$$ (7)

More generally, RCTC allows contraction over $k$ pairs of indices and is defined as

$$^{\ell_1+\ell_2-2k}\mathbf{T}_{a_1 \cdots a_{\ell_1-k} b_1 \cdots b_{\ell_2-k}} \\ = \frac{1}{\sqrt{3^k}} \sum_{i_1, \cdots, i_k} {}^{\ell_1}\mathbf{T}_{a_1 \cdots a_{\ell_1-k} i_1 \cdots i_k} \\ \cdot {}^{\ell_2}\mathbf{T}_{i_1 \cdots i_k b_1 \cdots b_{\ell_2-k}}.$$ (8)

The prefactor $1/\sqrt{3^k}$ is a normalization constant ensuring consistent scaling. The case $k = 0$ reduces to the RCTP, while $k > 0$ corresponds to the RCTC operation. We recommend interpreting RCTC as matrix multiplication in a suitably reshaped tensor shape. When Cartesian-$3j$ symbols are employed, the ICTP operation follows exactly the same computational steps as in spherical space, and its usage is fully consistent with that of e3nn (Thomas et al., 2018; Weiler et al., 2018; Kondor et al., 2018). For ICTC, a full implementation would require substantial modifications to e3nn, and therefore we do not provide this interface. However, if maintaining an identical interface is not required, both the

theoretical formulation and the corresponding code implementation are straightforward. To the best of our knowledge, no general formula has been found by us for decomposing the irreducible components resulting from tensor contraction. However, this decomposition is crucial for Cartesian models, where tensor contraction is indispensable. Owing to the exponentially increasing cost associated with Cartesian tensors, employing tensor contractions can provide notable gains in both efficiency and accuracy. Numerically, we can directly verify how the contracted irreducible tensors decompose into irreducible components. Since Cartesian tensors are generally considered to offer computational advantages only up to rank $\le 4$, we restrict the input ranks $\ell_1$ and $\ell_2$ to be at most 4. For completeness, we do not impose any limitation on the rank of the output tensor.

**Empirical Observations.** Through numerical experiments (see Table 7), we observe the following structural properties of the irreducible components obtained from tensor contractions:

1. For fixed $\ell_1$, $\ell_2$, and $k$, tensors associated with the same $\ell_3$ are linearly dependent.

2. For fixed $\ell_1$ and $\ell_2$ and varying $k$, tensors associated with the same $\ell_3$ are linearly independent.

3. For a given $(\ell_1, \ell_2, k)$, the range of irreducible components obtained from tensor contraction satisfies:

$$|\ell_1 - \ell_2| \le \ell_3 \le \ell_1 + \ell_2 - 2k.$$

### 4.4. Gate

To add nonlinearity to a tensor, there are essentially two distinct approaches. The first is to scale the entire tensor, and the second is to apply an invertible transformation that imposes nonlinearity. The treatment for Cartesian and spherical tensors is similar, so we do not elaborate here. In this work, when training within the proposed Cartesian framework, we find that nonlinearity is crucial for fitting large-scale datasets (OMat24, sAlex, and MPtrj) (Deng et al., 2023; Schmidt et al., 2024; Barroso-Luque et al., 2024). This is not an aspect that can be easily overlooked, and we will further investigate and discuss its impact in subsequent experiments. Here, we adopt the norm-based activation introduced in HotPP (Wang et al., 2024), which enables the incorporation of nonlinearity in an efficient manner. While it differs slightly from the existing implementation in e3nn, the overall concept is inherited from the former. Taking the SiLU activation as an example, its formulation is given as follows:

- **Scalar case** ($\ell = 0$):

$$\text{SiLU}(x) = \text{Sigmoid}(x) \cdot x.$$ (9)

- **Tensor case** ($\ell > 0$):

$$\mathrm{SiLU}(\mathbf{X}) = \mathrm{Sigmoid}\left(w \sum_{i=1}^{3^l}(X_i)^2 + b\right) \cdot \mathbf{X}, \quad (10)$$

where $w$ and $b$ are learnable parameters, $i$ indexes the tensor elements, and $\ell$ denotes both the rank and the weight.

# 5. Experiments

## 5.1. Conversion of spherical models into Cartesian ones

The 3BPA dataset (Kovács et al., 2021) is used as our primary benchmark, as it imposes strong requirements on extrapolation performance. This allows us to examine whether converting spherical models into Cartesian ones leads to any change in accuracy. We select three categories of models: the multi-layer message-passing model NequIP (Batzner et al., 2022), the ACE-based MACE (Batatia et al., 2022), and the strictly local, edge-features-based Allegro (Musaelian et al., 2023). It should be noted that the performance of different model architectures is not directly comparable. We did not tune hyperparameters to optimize accuracy, as our goal is solely to compare models with identical architectures and hyperparameters under identical training conditions. Apart from the possibility that the normalization of the Cartesian models may still be suboptimal, we train all models on the same hardware to ensure the fairest comparison possible. In this work, $L_{\max}$ and $\ell_{\max}$ respectively represent the truncation for node and edge.

**cNequIP & cAllegro.** From Table 1, we can observe that the accuracy remains nearly identical across almost all model sizes. However, for any given size, it is evident that the Cartesian model can not outperform the spherical model in terms of either computational speed or memory efficiency. In addition, for the edge-features-based Allegro model, which already has a large memory footprint, combining it with the Cartesian formulation further increases memory usage to the point that the model becomes impractical for real applications. From these observations, we conclude that the Cartesian models achieve accuracy comparable to their spherical counterparts, but they require dedicated architectural designs and should avoid edge-features-based constructions.

**cMACE.** From Table 2, we observe the same conclusion like the above models. In contrast to the previously discussed cNequIP and cAllegro models, cMACE further relies on Cartesian Generalized CG Coefficients, which introduces further limitations. Although its forward speed is indeed higher than that of TACE (Xu et al., 2026) and ICT potential (Zaverkin et al., 2024), which compute the product

basis from scratch, the use of precomputed product bases in Cartesian space comes with a critical drawback. The precomputed coefficients become extremely large, growing exponentially with both the correlation order and the tensor rank itself. As a result, the memory footprint increases dramatically once the correlation order $\geq 3$ and $\ell_{\max} \geq 3$.

## 5.2. Best Practices for Cartesian Models

From the previous experiments, we find that if one adopts architectures similar or identical to those used in spherical models, no improvement can be obtained. Therefore, we outline several important design principles for Cartesian models.

**Linear.** Schur's lemma states that only irreducible representations with the same weight can be linearly combined (Shao et al., 2025). However, due to shape inconsistency, Cartesian tensors that share the same weight but have different ranks can not be directly combined through linear transformations. Instead, they must first be converted into representations with matching weight and rank for such operations to be well defined. In principle, it is also possible to apply weighted combinations first and then perform the conversion across different ranks. However, lower-weight and highest-weight components are typically mixed, which is generally unfavorable for introducing learnable parameters. Another possible approach is to carry out linear transformations in the spherical basis and then convert the results back to the Cartesian basis.

**ICTP and ICTC.** Unlike spherical methods, Cartesian representations naturally support contraction operations. A key advantage of Cartesian models is that they can avoid the use of Cartesian-$3j$ symbols at the edge level, which is not possible in spherical methods. In the tensor product/contraction computations, three arrays are involved: node features lifted to edges, edge-level harmonics (spherical or Cartesian), and Wigner-$3j$ or Cartesian-$3j$ symbols. When interacting with neighboring atoms, Cartesian models can avoid edge-level Cartesian-$3j$, relying only on RCTP or RCTC. This is because Cartesian-$3j$ symbols act more like projection coefficients, rather than being strictly necessary for obtaining the separated tensorial form in spherical space. Retaining only the highest-weight components eliminates the need for learnable parameters for lower-weight components. Edge-level tensor products then require only two arrays in Cartesian space, which are then aggregated to node features. At the node level, a linear layer together with Cartesian-$3j$ symbols (which reduce to the ICTD matrices for the highest-weight components) is applied. This design improves both computational efficiency and memory usage. ICTC also ensures that lower-weight components are not completely discarded. Moreover, this design enables Carte-

*Table 1.* Root Mean Square Error (RMSE) for energies (E) [meV] and forces (F) [meV/Å] on the 3BPA dataset. The batches are divided into training and validation sets. Memory usage is reported in [MiB], and speed is reported in [s/epoch]. Model size naming follows the convention num_scalar(tensor)_channels-$L_{\max}$-$\ell_{\max}$-num_layers.

| Model Size | | 64(64)-2-2-4 | 64(64)-1-1-5 | 64(64)-2-2-5 | 64(64)-3-3-5 | 256(64)-2-2-3 | 256(64)-3-3-3 |
|---|---|---|---|---|---|---|---|
| Model Type | | NequIP/cNequIP | | | | Allegro/cAllegro | |
| 300K | E ($\downarrow$) | **5.0**/6.5 | **11.0**/**11.0** | **5.4**/5.8 | 4.9/**4.2** | 9.8/**7.9** | **6.6**/OOM |
| | F ($\downarrow$) | **16.9**/17.3 | 23.8/**23.4** | 16.5/**16.3** | **15.0**/**15.0** | 18.9/**18.1** | **16.2**/OOM |
| 600K | E ($\downarrow$) | **18.0**/20.6 | **26.1**/**26.1** | **17.3**/17.9 | 17.9/**15.7** | 17.0/**16.4** | **14.5**/OOM |
| | F ($\downarrow$) | **37.1**/37.6 | **53.0**/**53.0** | 37.0/**36.4** | 34.0/**33.8** | 42.3/**40.7** | **35.7**/OOM |
| 1200K | E ($\downarrow$) | **40.7**/42.9 | **64.5**/66.8 | 43.6/**42.2** | **38.4**/39.3 | 67.4/70.9 | **58.8**/OOM |
| | F ($\downarrow$) | **85.9**/88.6 | **130.8**/135.6 | 89.1/**89.0** | 85.4/**84.1** | 133.1/**132.3** | **119.3**/OOM |
| dihedral | E ($\downarrow$) | **12.0**/17.5 | 35.6/**18.2** | **15.9**/16.2 | 18.2/**15.1** | **32.0**/34.9 | **29.7**/OOM |
| | F ($\downarrow$) | 29.4/**26.8** | 44.2/**44.0** | 28.2/**26.3** | 27.6/**26.8** | 34.6/**33.6** | **33.2**/OOM |
| Speed | ($\downarrow$) | 3.1/3.1 | 3.0/3.0 | 4.8/4.8 | 15.8/46.4 | 2.6/2.7 | 2.8/OOM |
| Memory | ($\downarrow$) | 3342/4930 | 812/812 | 4294/6230 | 3968/12486 | 1792/3700 | 5862/OOM |
| Params | | 3.3M | 2.9M | 4.8M | 6.9M | 1.6M | 1.6M |
| Batch | | 5,25 | 5,5 | 5,25 | 5,5 | 5,5 | 5,5 |

[a] We did not test multiple random seeds as the purpose was merely to verify the correctness of our theory.

sian models to have exactly the same number of parameters as their spherical counterparts.

### 5.3. Universal Machine Learning Interatomic Potentials

Among a range of competitive and widely used uMLIPs evaluated on Matbench Discovery, Cartesian basis representations are rarely adopted. This is precisely because the irreducible model design theory for Cartesian bases has not been fully articulated before. Furthermore, with the exception of GRACE, which employs a two-layer architecture (Lysogorskiy et al., 2026), most representative uMLIPs adopt substantially deeper architectures, typically using at least six layers, with some incorporating convolutional modules spanning more than ten layers. As for MACE-MPA-0 (Batatia et al., 2025b), while it does not compete with the OAM series due to dataset limitations, we can still see from the thermal conductivity metric of MACE-OMAT-0-M that it does not achieve the best performance (despite its smaller parameter count, which is not the main reason). The GRACE model, owing to its well-designed product basis and the use of $\ell_{\max} = 4$ with correlation 4, has indeed achieved top-tier performance. When experimenting with our model, we initially used a 2-layer configuration: a 19M-parameter model with $L_{\max} = 2$, $\ell_{\max} = 3$, and tensor product paths constrained by $\ell_1 \leq \ell_2$, without gated interactions during convolution. However, from the thermal conductivity error in Table 3, we can observe that $\kappa$SRME for TACE-OMAT24-Linear is only 0.210. While better than MACE-OMAT-0-M and MACE-MH-1 (Batatia et al., 2025c), it still falls short of GRACE. In further tests, we continuously adjusted the architecture, but were unable to further improve the accuracy. Finally, we attempted to add some nonlinear modules during the convolution process. We

observed that both the model's accuracy and convergence speed significantly improved. This may not be immediately apparent, as seen in the BotNet (Batatia et al., 2025a), where removing the nonlinear module from NequIP does not affect its accuracy. Moreover, in MACE benchmarks (Batatia et al., 2022; Kovács et al., 2023), only linear interactions were used, which greatly improves extrapolation ability and accuracy for organic systems. In practice, we found that nonlinearity has a substantial impact on the model's performance for large datasets, although these metrics do not necessarily guarantee superior extrapolation ability. In our tests, the use of nonlinear interactions, such as on 3BPA, leads to a decrease in extrapolation ability, which contrasts with the results on large datasets.

**Performance.** The architecture we ultimately chose has $L_{\max} = 2$, $\ell_{\max} = 3$, with the restriction $\ell_1 \leq \ell_2$, 5 layers, correlation 2, and includes gate interactions. As summarized in Table 3, we successfully train TACE-v1-OMAT24-M and TACE-v1-OAM-M based on the OMat24, sAlex and MPtrj (OAM) datasets (Deng et al., 2023; Schmidt et al., 2024; Barroso-Luque et al., 2024) and these models demonstrate competitive performance across the evaluated benchmarks. It should be noted that we did not expect this family of models to achieve SOTA performance. In fact, this is still a relatively small model within the framework of uMLIP architectures. However, in Cartesian space, without an operator-fusion library, we can not use $L_{\max} = 4$ and $\ell_{\max} = 4$, as this would lead to a significant memory overhead.

**Speed.** In fact, under our architecture, when adopting a spherical basis, the Cartesian-basis variant can be faster. This is because our design avoids the use of Cartesian-

*Table 2.* Root Mean Square Error (RMSE) for energies (E) [meV] and forces (F) [meV/Å] on the 3BPA dataset. The batches are divided into training and validation sets. Memory usage is reported in [MiB], and speed is reported in [s/epoch]. Model size naming follows the convention num_channels-$L_{\max}$-$\ell_{\max}$-num_layers-correlation.

| Model Size | | 64-1-1-2-2 | 64-2-2-2-2 | 64-3-3-2-2 | 64-1-1-2-3 | 64-2-2-2-3 | 64-2-3-2-3 | 64-3-3-2-3 |
|---|---|---|---|---|---|---|---|---|
| Model Type | | MACE/cMACE | | | | | | |
| 300K | E(↓) | **7.3**/8.5 | **4.0/4.0** | 4.1/**3.5** | 9.1/**8.2** | **3.9**/4.5 | **3.6**/4.0 | **2.9**/3.6 |
| | F(↓) | 22.0/**21.9** | **12.2**/12.2 | **11.1**/11.4 | **21.0**/21.2 | **12.5/12.5** | **10.7**/11.3 | **10.1**/10.9 |
| 600K | E(↓) | **19.7**/19.8 | **11.4**/11.7 | 11.7/**11.5** | **18.6**/20.5 | **11.9**/12.9 | **11.5**/13.6 | **11.0**/12.0 |
| | F(↓) | **48.8**/49.1 | 28.5/**27.8** | **25.6**/27.1 | **46.7**/48.0 | **29.1**/29.3 | **25.6**/27.4 | **24.1**/26.0 |
| 1200K | E(↓) | 63.8/**62.4** | **33.5**/33.7 | **33.1**/36.4 | **61.4**/67.4 | **36.0**/38.4 | 40.8/**39.0** | **31.3**/37.9 |
| | F(↓) | 136.3/**136.1** | 79.0/**76.0** | **70.9**/82.3 | **132.8**/135.1 | **82.0**/82.5 | 85.0/**84.6** | **71.5**/80.2 |
| dihedral | E(↓) | 30.9/**29.0** | **8.1**/13.9 | 21.5/**16.6** | 34.2/**21.3** | 21.8/**12.1** | 18.5/**17.9** | **13.0**/24.1 |
| | F(↓) | 38.5/**35.8** | 21.8/**21.3** | 23.0/**19.4** | 36.1/**32.8** | 23.3/**24.6** | 20.2/**19.5** | **20.1**/26.3 |
| Speed | (↓) | 2.7/2.7 | 4.6/4.6 | 6.4/7.5 | 3.2/3.2 | 4.6/4.6 | 6.0/10.5 | 6.7/27.6 |
| Memory | (↓) | 846/846 | 1116/1402 | 1212/4542 | 862/862 | 1304/2038 | 1278/6408 | 1610/15890 |
| Params | | 0.64M | 1.2M | 2.2M | 0.65M | 1.2M | 1.8M | 2.2M |
| Batch | | 5,25 | 5,25 | 5,5 | 5,25 | 5,25 | 5,5 | 5,5 |

[a] We did not test multiple random seeds as the purpose was merely to verify the correctness of our theory.
[b] We removed MACE's dependency restriction on the e3nn version, since Cartesian Generalized CG Coefficients require higher-order Wigner-$3j$ symbols.

$3j$ symbols at the edge level, whereas the spherical formulation necessarily requires them. In terms of practical molecular dynamics speed comparisons, We select several representative models on Matbench Discovery, including NequIP-OAM-XL and MatRIS-10M-OAM. EquFlash is not included in this benchmark since it mainly serves as an acceleration architecture and shares a similar design philosophy with NequIP. In addition, we include MACE-OMAT-M as a representative model adopting $L_{\max} = 1$ and $\ell_{\max} = 3$, which provides a clear advantage in computational speed. However, it should be noted that its accuracy is comparatively lower; moreover, if reduced numerical precision were adopted, the computational speed of other models would also improve. As shown in Table 4, NequIP-AOTI encounters an out-of-memory (OOM) issue at 1536 atoms when using float64 precision. In contrast, despite the overhead introduced by the Cartesian basis, our model achieves a favorable balance between accuracy and efficiency. However, for NequIP, the use of operator fusion libraries, such as oeq (Bharadwaj et al., 2025), substantially reduces memory consumption and leads to significant speedups. Similarly, for MACE, the introduction of cueq (NVIDIA, 2024) also results in a marked improvement in computational efficiency. At present, however, comparable implementations are still lacking for the Cartesian space. Since NequIP does not explicitly provide an interface for precision conversion, we do not report results obtained with float32 precision.

## 6. Discussion, Limitations, and Outlook

This work develops a systematic framework for equivariant learning in Cartesian space within the context of atomistic modeling. Specifically, we have addressed the challenge of

*Table 3.* F1 and Thermal Conductivity Metrics ($\kappa$SRME) for a selection of representative uMLIPs.

| Model | F1(↑) | $\kappa$SRME (↓) | Params |
|---|---|---|---|
| TACE-v1-OAM-M | 0.889 | 0.173 | 18.8M |
| eSEN-30M-OAM | **0.925** | 0.170 | 30.2M |
| EquFlash | 0.919 | 0.158 | 28.7M |
| Nequip-OAM-XL | 0.906 | **0.125** | 32.1M |
| MatRIS-10M-OAM | 0.921 | 0.218 | 10.4M |
| Nequip-OAM-L | 0.893 | 0.166 | 9.6M |
| GRACE-2L-OAM-L | 0.883 | 0.169 | 26.4M |
| ORB v3 | 0.905 | 0.210 | 25.5M |
| SevenNet-MF-ompa | 0.901 | 0.317 | 25.7M |
| Allegro-OAM-L | 0.895 | 0.319 | 9.7M |
| GRACE-2L-OAM | 0.880 | 0.294 | 12.6M |
| DPA-3.1-3M-FT | 0.884 | 0.470 | 3.27M |
| MACE-MPA-0 | 0.852 | 0.412 | 9.06M |
| AlphaNet-v1-OMA | 0.901 | 0.644 | 4.65M |
| MatterSim v1 5M | 0.862 | 0.575 | 4.55M |
| TACE-v1-OMAT24-M | - | **0.158** | 18.8M |
| TACE-v1-OMAT24-Linear | - | 0.210 | 19.0M |
| GRACE-2L-OMAT-L-base | - | 0.165 | 26.4M |
| MACE-MH-1-OMAT-PBE | - | 0.228 | 6.4M |
| MACE-OMAT-M | - | 0.245 | 9.1M |

handling arbitrary irreducible components, an area where previous approaches fell short. We introduced Cartesian-$3j$ and Cartesian Generalized Clebsch-Gordan Coefficients, which serve as the foundation for defining the irreducible Cartesian tensor product and the irreducible Cartesian tensor contraction. These innovations enable the precise treatment of complex tensor operations in Cartesian space. We also propose the precomputation of Cartesian product bases and provide interfaces for transformations between Cartesian and spherical bases. These components allow us to

*Table 4.* This table reports the molecular dynamics simulation speed, measured in ps/day (↑), for several high-accuracy models using ASE (Larsen et al., 2017) as the simulation engine. TACE denotes TACE-v1-OAM-M, NequIP denotes NequIP-OAM-XL, MatRIS denotes MatRIS-10M-OAM, and MACE denotes MACE-OMAT-M. The comparison is performed using a single NVIDIA A800 80GB PCIe GPU.

| Atoms | Precision | TACE-torch | NequIP-AOTI | MatRIS-torch | NequIP-AOTI-oeq | MACE-torch | MACE-cueq |
|-------|-----------|------------|-------------|--------------|-----------------|------------|-----------|
| 192   | FP64      | 508        | **543**     | 43           | 1805            | 1385       | 1556      |
| 768   | FP64      | **174**    | 142         | 42           | 487             | 370        | 1002      |
| 1536  | FP64      | **85**     | OOM         | OOM          | 248             | 226        | 735       |
| 2304  | FP64      | OOM        | OOM         | OOM          | 146             | 84         | 324       |
| 3456  | FP64      | OOM        | OOM         | OOM          | 99              | OOM        | 180       |
| 2304  | FP32      | 80         | -           | -            | -               | 194        | 501       |
| 3456  | FP32      | 53         | -           | -            | -               | 118        | 287       |

[a] Due to policy restrictions, we only compared against models that were publicly accessible to us. For fairness, the **boldfaced values** correspond to models implemented with pure Torch, while models using acceleration libraries are included only for reference. Note that the MACE-OMAT-M are not included in Matbench Discovery and are therefore provided here only as additional references.

instantiate Cartesian counterparts of common spherical architectures and to run controlled experiments in which the architecture is fixed and only the representation basis is changed.

Our controlled comparisons show that irreducible Cartesian models can achieve accuracy comparable to their spherical counterparts, but direct Cartesianization incurs unfavorable compute and memory scaling due to the rapid dimensional growth of Cartesian tensors. These results support the central conclusion of the paper: Cartesian tensor formulations are viable for equivariant MLIPs, but their practical use requires dedicated architectural choices rather than a drop-in basis substitution. As a proof of concept, we carefully design a Cartesian-framework model architecture, under which the uMLIP model trained with the proposed framework achieves competitive performance on Matbench Discovery. These results indicate that irreducible Cartesian constructions are not only theoretically sound but also practical for universal atomistic models. We expect the ICT primitives to enable further development of efficient Cartesian-based models and Cartesian-spherical hybrids.

The Cartesian approach also faces inherent limitations. The dimensionality curse restricts the use of very high truncations within the Cartesian model. Although, the accuracy of the Cartesian model is comparable to that of spherical models, the spherical methods can take advantage of higher truncations as computational resources improve. In comparison with CGTP libraries such as e3nn, cueq, and oeq (Thomas et al., 2018; Weiler et al., 2018; Kondor et al., 2018; NVIDIA, 2024; Bharadwaj et al., 2025), the current pure PyTorch (Paszke et al., 2019) implementation of our Cartesian method still exhibits higher memory usage. This is mainly attributed to the exponential growth in the number of Cartesian tensor elements and the lack of operator fusion and specialized kernels in the Cartesian approach.

Several aspects of the framework remain open and motivate follow-up work. First, efficient sparse storage of higher-rank tensors in Cartesian space and strategies for selecting sparse paths are important for scaling. Second, extending the framework to additional symmetry-aware constructions (e.g., SO(2) group). Third, an interesting direction is the hybrid approaches that integrate Cartesian and spherical methods, leveraging complementary strengths of each representation, and has the potential of yielding performance or efficiency gains beyond what either basis achieves in their own.

## Code Availability

We have released the implementations, including Cartesian-$3j$ symbols, generalized Clebsch-Gordan coefficients, Cartesian harmonics, precomputed product bases, ICTD, and ICTP, at https://github.com/xvzemin/cartnn, https://github.com/xvzemin/tace, https://github.com/xvzemin/cartesian_nequip, https://github.com/xvzemin/cartesian_allegro, and https://github.com/xvzemin/cartesian_mace.

## Author contribution

W.X. and P.H. conceived the project and guided the research. The code, equations, and tables were prepared by Z.X., while C.W. contributed to the benchmark results. All authors edited and revised the manuscript.

## Acknowledgment

This work was supported by the National Natural Science Foundation of China NSFC (22433004 and 22403064), the open research fund of Key Laboratory of Precision and Intelligent Chemistry, and ShanghaiTech University. We are also grateful for the computing time provided by the the HPC Platform of ShanghaiTech University.

## Impact Statement

This paper opens a new avenue for atomistic modeling by introducing a systematic irreducible Cartesian tensor framework. This expands the design space beyond the prevailing spherical-tensor paradigm, which has the potential to accelerate scientific discovery in chemistry and material science. Due to the generic nature of pure science, none of which we feel must be specifically highlighted here.

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

## A. Cartesian-$3j$

The second interpretation of ICTP can be understood as follows: we first form the RCTP of the inputs, then apply the basis change to extract the corresponding irreducible components in spherical space. Next, we perform a basis transformation back to Cartesian space so that all irreducible components satisfy weight = rank. Thus, the Cartesian-$3j$ can be written as:

$$Z(\ell_1, \ell_2, \ell_3) = {}^{(\ell_1+\ell_2;\ell_3;1)}C {}^{(\ell_3;\ell_3;1)}C^T. \tag{11}$$

We find that after performing the RCTP, Cartesian tensors of weight $l_3$ (nonzero element) may contain more than one multiplicity. However, after converting them to spherical tensors, they turn out to be linearly dependent. Therefore, following the parentage scheme order, we select the first occurrence of $\ell_3$, which differs by at most a constant factor. Note that this is not numerically identical to Eq. (5); however, the resulting ICTs are the same.

*Table 5.* Root Mean Square Error (RMSE) for energies (E) [meV] and forces (F) [meV/Å] on the 3BPA dataset. The batches are divided into training and validation sets. Memory usage is reported in [MiB], and speed is reported in [s/epoch]. Model size naming follows the convention num_scalar(tensor)_channels-$L_{max}$-$\ell_{max}$-num_layers.

| Model Size | | 64(64)-2-2-4 | 64(64)-1-1-5 | 64(64)-2-2-5 | 64(64)-3-3-5 | 256(64)-2-2-3 | 256(64)-3-3-3 |
|---|---|---|---|---|---|---|---|
| Model Type | | | | NequIP/cNequIP | | Allegro/cAllegro | |
| 300K | E (↓) | 5.0/5.6 | 11.0/9.4 | 5.4/4.8 | 4.9/5.2 | 9.8/9.1 | 6.6/OOM |
| | F (↓) | 16.9/17.2 | 23.8/23.9 | 16.5/16.6 | 15.0/15.4 | 18.9/18.3 | 16.2/OOM |
| 600K | E (↓) | 18.0/19.5 | 26.1/26.1 | 17.3/18.7 | 17.9/18.0 | 17.0/17.0 | 14.5/OOM |
| | F (↓) | 37.1/37.7 | 53.0/54.5 | 37.0/37.1 | 34.0/34.5 | 42.3/41.3 | 35.7/OOM |
| 1200K | E (↓) | 40.7/42.0 | 64.5/67.8 | 43.6/42.7 | 38.4/38.2 | 67.4/69.4 | 58.8/OOM |
| | F (↓) | 85.9/88.3 | 130.8/137.0 | 89.1/90.0 | 85.4/84.7 | 133.1/132.0 | 119.3/OOM |
| dihedral | E (↓) | 12.0/11.8 | 35.6/18.2 | 15.9/14.3 | 18.2/18.7 | 32.0/35.6 | 29.7/OOM |
| | F (↓) | 29.4/29.5 | 44.2/44.4 | 28.2/28.8 | 27.6/28.8 | 34.6/32.9 | 33.2/OOM |
| Speed | (↓) | 3.1/3.1 | 3.0/3.0 | 4.8/4.8 | 15.8/46.4 | 2.6/2.7 | 2.8/OOM |
| Memory | (↓) | 3342/4930 | 812/812 | 4294/6230 | 3968/12486 | 1792/3700 | 5862/OOM |
| Params | | 3.3M | 2.9M | 4.8M | 6.9M | 1.6M | 1.6M |
| Batch | | 5,25 | 5,5 | 5,25 | 5,5 | 5,5 | 5,5 |

[a] We did not test multiple random seeds as the purpose was merely to verify the correctness of our theory.

*Table 6.* Root Mean Square Error (RMSE) for energies (E) [meV] and forces (F) [meV/Å] on the 3BPA dataset. The batches are divided into training and validation sets. Memory usage is reported in [MiB], and speed is reported in [s/epoch]. Model size naming follows the convention num_channels-$L_{max}$-$\ell_{max}$-num_layers-correlation.

| Model Size | | 64-1-1-2-2 | 64-2-2-2-2 | 64-3-3-2-2 | 64-1-1-2-3 | 64-2-2-2-3 | 64-2-3-2-3 | 64-3-3-2-3 |
|---|---|---|---|---|---|---|---|---|
| Model Type | | | | | MACE/cMACE | | | |
| 300K | E(↓) | 7.3/10.5 | 4.0/4.1 | 4.1/3.3 | 9.1/8.3 | 3.9/4.3 | 3.6/3.0 | 2.9/3.2 |
| | F(↓) | 22.0/21.4 | 12.2/13.3 | 11.1/11.9 | 21.0/21.6 | 12.5/12.9 | 10.7/10.9 | 10.1/11.1 |
| 600K | E(↓) | 19.7/21.4 | 11.4/13.5 | 11.7/13.2 | 18.6/20.5 | 11.9/12.3 | 11.5/13.3 | 11.0/11.0 |
| | F(↓) | 48.8/48.1 | 28.5/30.0 | 25.6/28.1 | 46.7/49.0 | 29.1/29.6 | 25.6/26.8 | 24.1/26.5 |
| 1200K | E(↓) | 63.8/62.3 | 33.5/37.3 | 33.1/36.1 | 61.4/67.5 | 36.0/39.2 | 40.8/37.0 | 31.3/39.2 |
| | F(↓) | 136.3/131.8 | 79.0/80.6 | 70.9/82.0 | 132.8/137.9 | 82.0/83.1 | 85.0/80.6 | 71.5/81.5 |
| dihedral | E(↓) | 30.9/17.5 | 8.1/10.7 | 21.5/13.6 | 34.2/28.8 | 21.8/12.1 | 18.5/12.8 | 13.0/19.0 |
| | F(↓) | 38.5/36.2 | 21.8/22.2 | 23.0/20.4 | 36.1/36.2 | 23.3/26.0 | 20.2/21.1 | 20.1/22.4 |
| Speed | (↓) | 2.7/2.7 | 4.6/4.6 | 6.4/7.5 | 3.2/3.2 | 4.6/4.6 | 6.0/10.5 | 6.7/27.6 |
| Memory | (↓) | 846/846 | 1116/1402 | 1212/4542 | 862/862 | 1304/2038 | 1278/6408 | 1610/15890 |
| Params | | 0.64M | 1.2M | 2.2M | 0.65M | 1.2M | 1.8M | 2.2M |
| Batch | | 5,25 | 5,25 | 5,5 | 5,25 | 5,25 | 5,5 | 5,5 |

[a] We did not test multiple random seeds as the purpose was merely to verify the correctness of our theory.
[b] We removed MACE's dependency restriction on the e3nn version, since Cartesian Generalized CG Coefficients require higher-order Wigner-$3j$ symbols.

## B. Numerical Experiments

After performing the reducible Cartesian tensor contractions, we verify that, during the transformation between the spherical and Cartesian representations, the zero-valued components correspond to irreducible representations that do not appear in the decomposition. The multiplicities associated with the nonzero components can be greater than one. Whether these components are linearly independent or linearly dependent depends on the specific combination of $(\ell_1, \ell_2, k)$.

*Table 7.* The irreducible representation components produced by irreducible tensor contraction.

| $\ell_1$ | $\ell_2$ | $k$ | Irreducible components $\ell_3$ | multiplicity for each $\ell_3$ |
|---|---|---|---|---|
| 1 | 1 | 1 | 0 | [1] |
| 1 | 2 | 1 | 1 | [1] |
| 1 | 3 | 1 | 2 | [1] |
| 1 | 4 | 1 | 3 | [1] |
| 2 | 1 | 1 | 1 | [1] |
| 2 | 2 | 2 | 0 | [1] |
| 2 | 2 | 1 | $0 \oplus 1 \oplus 2$ | $[1, 1, 1]$ |
| 2 | 3 | 2 | 1 | [1] |
| 2 | 3 | 1 | $1 \oplus 2 \oplus 3$ | $[1, 1, 1]$ |
| 2 | 4 | 2 | 2 | [1] |
| 2 | 4 | 1 | $2 \oplus 3 \oplus 4$ | $[1, 1, 1]$ |
| 3 | 1 | 1 | 2 | [1] |
| 3 | 2 | 2 | 1 | [1] |
| 3 | 2 | 1 | $1 \oplus 2 \oplus 3$ | $[3, 2, 1]$ |
| 3 | 3 | 3 | 0 | [1] |
| 3 | 3 | 2 | $0 \oplus 1 \oplus 2$ | $[1, 1, 1]$ |
| 3 | 3 | 1 | $0 \oplus 1 \oplus 2 \oplus 3 \oplus 4$ | $[1, 2, 3, 2, 1]$ |
| 3 | 4 | 3 | 1 | [1] |
| 3 | 4 | 2 | $1 \oplus 2 \oplus 3$ | $[1, 1, 1]$ |
| 3 | 4 | 1 | $1 \oplus 2 \oplus 3 \oplus 4 \oplus 5$ | $[1, 2, 3, 2, 1]$ |
| 4 | 1 | 1 | 3 | [1] |
| 4 | 2 | 2 | 2 | [1] |
| 4 | 2 | 1 | $2 \oplus 3 \oplus 4$ | $[6, 3, 1]$ |
| 4 | 3 | 3 | 1 | [1] |
| 4 | 3 | 2 | $1 \oplus 2 \oplus 3$ | $[1, 1, 1]$ |
| 4 | 3 | 1 | $1 \oplus 2 \oplus 3 \oplus 4 \oplus 5$ | $[6, 7, 6, 3, 1]$ |
| 4 | 4 | 4 | 0 | [1] |
| 4 | 4 | 3 | $0 \oplus 1 \oplus 2$ | $[1, 1, 1]$ |
| 4 | 4 | 2 | $0 \oplus 1 \oplus 2 \oplus 3 \oplus 4$ | $[1, 2, 3, 2, 1]$ |
| 4 | 4 | 1 | $0 \oplus 1 \oplus 2 \oplus 3 \oplus 4 \oplus 5 \oplus 6$ | $[1, 3, 6, 7, 6, 3, 1]$ |
| ... | ... | ... | ... | ... |

## C. Model Architecture

This section describes the detailed architecture of our model. The chemical element information is initialized as scalar node features, corresponding to irreducible representations with $\ell = 0$. Higher-order geometric information is introduced progressively through multiple equivariant interaction and product layers. At each layer, the feature of central atoms (target) are updated by aggregating information from its neighboring atoms (source) within a cutoff radius. The angular information on the edges is encoded by expanding the unit vectors between the central atom and its neighboring atoms using Cartesian harmonics. The two-body radial information is represented by expanding the interatomic distances with zeroth-order spherical Bessel functions. In addition, the chemical element types of the central and neighboring atoms are encoded using one-hot representations and subsequently transformed through linear layers. The radial features and the embedded element information are then concatenated and passed through a multilayer perceptron (MLP) with layer normalization. The output of this MLP is used to generate the convolution weights. The interaction between neighboring atomic features and Cartesian harmonics is implemented through highest weight ICTP and ICTC. After the equivariant convolution, we adopt the Atomic

Cluster Expansion (ACE) framework (Drautz, 2019) to construct expressive atomic representations. Linear layers can be chosen to be either element-aware or element-agnostic. After the atomic basis functions are obtained, higher-order correlations are captured by constructing product bases. These product bases are formed also using highest weight ICTP and ICTC between atomic basis. Through the combination of equivariant message passing and explicit construction of many-body information, the model is able to efficiently encode both local geometric environments in a symmetry-consistent manner. A particularly important point to note is that the gate component has a significant impact on the model performance. In our tests, the choice of normalization after convolution, using a density like (Batatia et al., 2025c) in Eq 18 scheme or using a statistically estimated average number of neighboring atoms, does not lead to a substantial difference in performance. Nevertheless, we adopt the density like normalization throughout this work. In the following formulas, $l_1$ and $l_4$ belong to $L_{\max}$, while $l_2$ and $l_3$ belong to $l_{\max}$. Here, $\mathbf{h}$ denotes the node features, $\phi$ represents the particle basis, $\mathbf{A}$ denotes the atomic basis, $\mathbf{B}$ denotes the product basis, and $R$ represents the convolution weights. When treated as variables, $z_i$ and $z_j$ denote one hot encodings followed by a linear layer. The index $i$ refers to the target atom, and $j$ refers to the source atom.

$$f_{\text{cut}}(r_{ij}) = \begin{cases} 1 - \dfrac{(p+1)(p+2)}{2}\left(\dfrac{r_{ij}}{c}\right)^p + p(p+2)\left(\dfrac{r_{ij}}{c}\right)^{p+1} - \dfrac{p(p+1)}{2}\left(\dfrac{r_{ij}}{c}\right)^{p+2}, & 0 \le r_{ij} \le c, \\ 0, & r_{ij} > c, \end{cases} \tag{12}$$

$$j_0^n(r_{ij}) = \sqrt{\frac{2}{c}}\frac{\sin(\frac{n\pi}{c}r_{ij})}{r_{ij}}, \tag{13}$$

$$R^{(t)}_{p_1 c\, l_1 l_2 l_3 z_i z_j}(r_{ij}) = \text{MLP}\left(j_0^n(r_{ij}), z_i, z_j\right) f_{\text{cut}}(r_{ij}), \tag{14}$$

$$^{(l_2;l_2;1)}\mathbf{E}_{ij} = \frac{(2l_2-1)!!}{l_2!} \cdot {}^{(l_2;l_2;1)}\mathcal{T}\left(\underbrace{\hat{\mathbf{r}}_{ij} \otimes \cdots \otimes \hat{\mathbf{r}}_{ij}}_{l_2 \text{ times}}\right), \tag{15}$$

$$^{l_3}_{p_1,c}\phi^{(t)}_{ij} = R^{(t)}_{p_1 c l_1 l_2 l_3 z_i z_j}(r_{ij}) \cdot \left(^{l_1}_c\mathbf{h}^{(t)}_i \otimes {}^{(l_2;l_2;1)}\mathbf{E}_{ij}\right), \tag{16}$$

$$^{l_3}_c\mathbf{A}^{(t)}_i = \frac{1}{\mathcal{N}(i)} \cdot {}^{(l_3;l_3;1)}\mathcal{T}\left(\sum_{p_1,\tilde{c}} W^{(t)}_{p_1 \tilde{c} l_3 c} \sum_{j\in\mathcal{N}(i)} {}^{l_3}_{p_1,c}\phi^{(t)}_{ij}\right), \tag{17}$$

$$\mathcal{N}(i) = \beta \sum_{j\in\mathcal{N}(i)} \tanh\left(\left[\text{MLP}(j_0^n(r_{ij}), z_i, z_j)\right]^2 f_{\text{cut}}(r_{ij})\right) + \alpha, \tag{18}$$

$$^{l_3}_c\mathbf{A}^{(t)}_i = \begin{cases} \sum_{\tilde{c}} W^{(t)}_{l_3\tilde{c}c}\,\text{NormGate}\left(^{l_3}_c\mathbf{A}^{(t)}_i\right), & t=1, \\ \sum_{\tilde{c}} W^{(t)}_{l_3\tilde{c}c}\,\text{NormGate}\left(^{l_3}_c\mathbf{A}^{(t)}_i\right) + \sum_{\tilde{c}} W^{(t)}_{z_i l_3\tilde{c}c}\,{}^{l_3}_c\mathbf{h}^{(t-1)}_i, & t>1. \end{cases} \tag{19}$$

$$^{l_4}_{p_2 c}\mathbf{B}^{(t)}_{i,\nu} = \underbrace{\bigotimes_{\xi=1}^{\nu} {}^{(l_3;\xi)}_c\mathbf{A}^{(t)}_i}_{\text{with } {}^{(l_4;l_4;1)}\mathcal{T}}, \tag{20}$$

$$^{l_4}_c\mathbf{h}^{(t+1)}_i = \begin{cases} \sum_{\nu p_2 c} W^{(t)}_{\nu p_2 c z_i l_4}\,{}^{l_4}_{p_2 c}\mathbf{B}^{(t)}_{i,\nu} & t=1, \\ \sum_{\nu p_2 c} W^{(t)}_{\nu p_2 c z_i l_4}\,{}^{l_4}_{p_2 c}\mathbf{B}^{(t)}_{i,\nu} + \sum_{\tilde{c}} W^{(t)}_{z_i l_4\tilde{c}c}\,{}^{l_4}_c\mathbf{A}^{(t)}_i & t>1, \end{cases} \tag{21}$$

$$\mathcal{R}^{(t)}\begin{pmatrix}0\\c h_i^{(t)}\end{pmatrix} = \begin{cases} \sum_c W_c^{(t)} {}_c^0 h_i^{(t)} & t < T, \\ \text{MLP}\begin{pmatrix}0\\c h_i^{(t)}\end{pmatrix} & t = T, \end{cases} \tag{22}$$

$$\text{E}_i = \text{E}_{z_i} + \sum_{t=1}^{T} \text{E}_i^{(t)} = \text{E}_{z_i} + \sum_{t=1}^{T} \mathcal{R}^{(t)}\begin{pmatrix}0\\c h_i^{(t)}\end{pmatrix}. \tag{23}$$

## D. Training Details

For the 3BPA benchmarks, we use a single NVIDIA RTX 4090 GPU together with an AMD EPYC 9554 64-Core Processor (64 cores / 128 threads) and train the model in float32 precision. For uMLIPs training, we use eight NVIDIA H100 GPUs. The total training time for the OAM and OMAT models is approximately 960 hours. All models are trained using tf32 precision. The sAlex dataset together with eight splits of the MPtrj dataset is trained for two epochs. The loss function is the Huber loss with $\delta = 0.01$. During pretraining, the weights for per atom energy, forces, and stress are set to a ratio of $1 : 8 : 8$. During finetuning, the loss weights are changed to $1 : 1 : 2$, and stochastic weight averaging is applied. In addition, the comparison of uMLIPs inference speed was performed using a single NVIDIA A800 80GB PCIe GPU together with dual Intel Xeon Silver 4314 CPUs (32 cores / 64 threads in total).

*Table 8.* Hyper-parameters for training on the OMat24, sAlex, and MPTrj datasets.

| Hyper-parameters | OMAT24 | sAlex+MPtrj |
|---|---|---|
| Optimizer | AdamW | AdamW |
| Learning rate scheduling | CosineAnnealingWarmupRestarts | None |
| Warmup ratio | 5% | 0% |
| Stochastic Weight Averaging | False | True |
| Maximum learning rate | $4 \times 10^{-3}$ | $1 \times 10^{-4}$ |
| Minimum learning rate | $5 \times 10^{-4}$ | $1 \times 10^{-4}$ |
| Batch size | 512 | 256 |
| Number of epochs | 4 | 2 |
| Weight decay | $1 \times 10^{-8}$ | $1 \times 10^{-8}$ |
| Loss huber delta: | 0.01 | 0.01 |
| Energy coefficient | 1 | 1 |
| Force coefficient | 8 | 1 |
| Stress coefficient | 8 | 2 |
| Gradient clipping norm threshold | 0.25 | 0.25 |
| Model EMA decay | 0.995 | 0.995 |
| Cutoff radius ($\mathring{A}$) | 6 | 6 |
| Number of layers | 5 | 5 |
| Number of channels | 48 | 48 |
| $L_{\max}$ | 2 | 2 |
| $\ell_{\max}$ | 3 | 3 |
| $\ell_1 \ell_2$ | $\leq$ | $\leq$ |
| Correlation order | 2 | 2 |

## E. Different Tensor Product Strategies

At present, there exist many different approaches for performing tensor products. In particular, tensor product strategies in spherical spaces have been systematically summarized in (Xie et al., 2025). Here, we provide a brief discussion by considering both spherical and Cartesian spaces within the SO(3) group. CGTP and highest weight ICTP/ICTC possess exactly the same number of parameters. Therefore, any model based on CGTP can, in principle, be converted into a Cartesian formulation. Moreover, CGTP and ICTP/ICTC may be regarded as the gold standard for tensor products, since

they allow all possible paths and assign independent weights to each path, thereby providing the strongest expressive power. In contrast, Gaunt Tensor Product (GTP) (Luo et al., 2024) and its Cartesian counterpart introduce equivariance errors and do not explicitly contain the concept of paths. In addition, both lack antisymmetric interactions. Matrix Tensor Product (MTP) (Unke & Maennel, 2024) transforms tensor products into matrix multiplications by contracting spherical tensors with CG coefficients into matrices. Similarly, MTP does not explicitly contain the concept of paths, but it does include antisymmetric interactions. Since ICTC can also be reformulated as matrix multiplication after reshaping, MTP and ICTC may be equivalent at low tensor ranks, although this requires further investigation.

## F. Matbench Discovery

*Table 9.* Unique Prototypes entries in Matbench Discovery at the time of uploading TACE-v1-OAM-M.

| Model | CPS | Acc | F1 | DAF | Prec | MAE | $R^2$ | $\kappa$SRME | RMSD | Params | Date Added |
|---|---|---|---|---|---|---|---|---|---|---|---|
| eSEN-30M-OAM | 0.888 | 0.977 | 0.925 | 6.069 | 0.928 | 0.018 | 0.866 | 0.170 | 0.061 | 30.2M | 2025-03-17 |
| EquFlash | 0.888 | 0.975 | 0.919 | 5.983 | 0.915 | 0.019 | 0.871 | 0.158 | 0.060 | 28.7M | 2025-06-23 |
| Nequip-OAM-XL | 0.886 | 0.971 | 0.906 | 5.869 | 0.897 | 0.020 | 0.872 | 0.125 | 0.063 | 32.1M | 2025-11-30 |
| MatRIS-10M-OAM | 0.877 | 0.976 | 0.921 | 6.039 | 0.923 | 0.019 | 0.871 | 0.218 | 0.060 | 10.4M | 2025-10-29 |
| Nequip-OAM-L | 0.870 | 0.967 | 0.893 | 5.823 | 0.890 | 0.022 | 0.865 | 0.166 | 0.065 | 9.6M | 2025-09-08 |
| TACE-v1-OAM-M | 0.867 | 0.965 | 0.889 | 5.749 | 0.879 | 0.022 | 0.865 | 0.173 | 0.065 | 18.8M | 2026-01-06 |
| GRACE-2L-OAM-L | 0.865 | 0.964 | 0.883 | 5.840 | 0.893 | 0.022 | 0.862 | 0.169 | 0.064 | 26.4M | 2025-09-09 |
| ORB v3 | 0.860 | 0.971 | 0.905 | 5.912 | 0.904 | 0.024 | 0.821 | 0.210 | 0.075 | 25.5M | 2025-04-05 |
| Allegro-OAM-L | 0.840 | 0.966 | 0.895 | 5.674 | 0.867 | 0.022 | 0.868 | 0.319 | 0.065 | 9.7M | 2025-09-08 |
| GRACE-2L-OAM | 0.837 | 0.963 | 0.880 | 5.774 | 0.883 | 0.023 | 0.862 | 0.294 | 0.067 | 12.6M | 2025-02-06 |
| DPA-3.1-3M-FT | 0.802 | 0.963 | 0.884 | 5.667 | 0.866 | 0.023 | 0.869 | 0.470 | 0.069 | 3.27M | 2025-06-05 |
| eSEN-30M-MP | 0.797 | 0.946 | 0.831 | 5.260 | 0.804 | 0.033 | 0.822 | 0.340 | 0.075 | 30.1M | 2025-03-17 |
| MACE-MPA-0 | 0.795 | 0.954 | 0.852 | 5.582 | 0.853 | 0.028 | 0.842 | 0.412 | 0.073 | 9.06M | 2024-12-09 |
| MatRIS-10M-MP | 0.778 | 0.951 | 0.847 | 5.422 | 0.829 | 0.031 | 0.824 | 0.489 | 0.072 | 10.4M | 2025-10-29 |
| AlphaNet-v1-OAM | 0.769 | 0.968 | 0.901 | 5.747 | 0.879 | 0.024 | 0.831 | 0.644 | 0.079 | 4.65M | 2025-05-12 |
| MatterSim v1 5M | 0.767 | 0.959 | 0.862 | 5.852 | 0.895 | 0.024 | 0.863 | 0.575 | 0.073 | 4.55M | 2024-12-16 |
| GRACE-1L-OAM | 0.761 | 0.944 | 0.824 | 5.255 | 0.803 | 0.031 | 0.842 | 0.517 | 0.072 | 3.45M | 2025-02-06 |
| Eqnorm MPtrj | 0.756 | 0.929 | 0.786 | 4.844 | 0.741 | 0.040 | 0.799 | 0.408 | 0.084 | 1.31M | 2025-05-26 |
| Nequip-MP-L | 0.733 | 0.921 | 0.761 | 4.704 | 0.719 | 0.043 | 0.791 | 0.452 | 0.086 | 9.6M | 2025-09-08 |
| Nequix MP | 0.729 | 0.914 | 0.751 | 4.455 | 0.681 | 0.044 | 0.782 | 0.446 | 0.085 | 708k | 2025-08-17 |
| Allegro-MP-L | 0.720 | 0.915 | 0.751 | 4.516 | 0.690 | 0.044 | 0.778 | 0.504 | 0.082 | 18.7M | 2025-09-08 |
| DPA-3.1-MPtrj | 0.718 | 0.936 | 0.803 | 5.024 | 0.768 | 0.037 | 0.812 | 0.650 | 0.080 | 4.81M | 2025-06-05 |
| SevenNet-l3i5 | 0.714 | 0.920 | 0.760 | 4.629 | 0.708 | 0.044 | 0.776 | 0.550 | 0.085 | 1.17M | 2024-12-10 |
| HIENet | 0.707 | 0.929 | 0.777 | 4.932 | 0.754 | 0.041 | 0.793 | 0.642 | 0.080 | 7.51M | 2025-07-01 |
| GRACE-2L-MPtrj | 0.681 | 0.895 | 0.691 | 4.163 | 0.636 | 0.052 | 0.741 | 0.526 | 0.090 | 15.3M | 2024-11-21 |
| MACE-MP-0 | 0.637 | 0.878 | 0.669 | 3.777 | 0.577 | 0.057 | 0.697 | 0.682 | 0.092 | 4.69M | 2023-07-14 |
| eqV2 M | 0.558 | 0.975 | 0.917 | 6.047 | 0.924 | 0.020 | 0.848 | 1.771 | 0.069 | 86.6M | 2024-10-18 |
| ORB v2 | 0.528 | 0.965 | 0.880 | 6.041 | 0.924 | 0.028 | 0.824 | 1.734 | 0.097 | 25.2M | 2024-10-11 |
| eqV2 S DeNS | 0.522 | 0.939 | 0.815 | 5.042 | 0.771 | 0.036 | 0.788 | 1.676 | 0.076 | 31.2M | 2024-10-18 |
| ORB v2 MPtrj | 0.470 | 0.922 | 0.765 | 4.702 | 0.719 | 0.045 | 0.756 | 1.726 | 0.101 | 25.2M | 2024-10-14 |
| CHGNet | 0.343 | 0.851 | 0.613 | 3.361 | 0.514 | 0.063 | 0.689 | 2.000 | 0.095 | 413k | 2023-03-03 |
| M3GNet | 0.310 | 0.812 | 0.569 | 2.882 | 0.441 | 0.075 | 0.585 | 2.000 | 0.112 | 228k | 2022-09-20 |
| GNoME | n/a | 0.948 | 0.829 | 5.523 | 0.844 | 0.035 | 0.785 | n/a | n/a | 16.2M | 2024-02-03 |

