# OpenReview forum: "A Cartesian-3j Framework for Machine Learning Interatomic Potentials"
_ICML.cc/2026/Conference — ICML 2026 regular_

### Official Review · Reviewer_GSPB · 2026-03-09

**Soundness:** 3
**Presentation:** 2
**Significance:** 3
**Originality:** 3
**Overall Recommendation:** 3
**Confidence:** 3

**Summary:**

This paper proposes a representation method for Machine Learning Interatomic Potential (MLIP) models based on Irreducible Cartesian Tensor (ICT). The study aims to explore the feasibility, advantages, and challenges of ICT-based representations. While existing models predominantly employ Spherical Tensor (ST) representations, research on ICT-based alternatives remains limited. The proposed ICT method achieves two key innovations: (1) a theoretical framework proposes Cartesian-3j and Cartesian-nj symbols—Cartesian analogs of the Wigner symbols used in ST methods—which provide the mathematical foundation for product and contraction operations on ICT; and (2) an extension of the e3nn library to support ICT operations, enabling the construction of Cartesian versions of mainstream models such as NequIP (e.g., cNequIP).

Experiments are conducted under strictly controlled network architectures and hyperparameters to compare spherical-harmonic-based models (ST) and Cartesian-based models (ICT). Results show that Cartesian-based models achieve accuracy comparable to spherical-harmonic-based models on datasets such as OMat24, but at the cost of higher computational and memory demands. The analysis further reveals the relative advantages and disadvantages of ST and ICT methods in representing linear transformations, tensor products, and tensor contractions. Additionally, an optimized Cartesian-based model is designed and shown to deliver competitive performance.

**Compliance With Llm Reviewing Policy:**

Affirmed.

**Final Justification:**

The authors' response effectively addressed the main issues I raised; therefore, I increase the Soundness score by 1 point. However, considering the overall contributions and limitations of the paper, I keep the Overall Recommendation unchanged from my original assessment.

**Key Questions For Authors:**

1. It is recommended to supplement Tables 1 and 2 with additional accuracy metrics—such as Mean Absolute Error and Coefficient of Determination—to enable a more comprehensive evaluation of model performance.

2. The missing F1 metric results in Table 3 should be provided to complete the evaluation.

3. The “Related Work” section should be expanded at its end to clearly articulate the core challenges and innovations that underpin the ICT method.

4. Optimal performance values in Tables 1-4 should be highlighted (e.g., in bold) to facilitate comparison.

5. The manuscript should be reviewed for consistent and proper use of abbreviations throughout, with all terms defined upon first appearance.

**Limitations:**

yes

**Strengths And Weaknesses:**

**Strengths**
1. Through theoretical innovation (proposes Cartesian-3j and Cartesian-nj symbols) and engineering implementation (extension of the e3nn library), the study constructs Cartesian-based versions of several mainstream MLIP models. This establishes an experimental foundation for comparative studies of ST and ICT methods and provides an empirical analysis of their respective strengths and weaknesses in representing linear transformations, tensor products, and tensor contractions.

2. Extensive experiments validate the feasibility and potential of ICT-based representations, offering a new direction for MLIP representation paradigms.

**Weaknesses**
1. Tables 1 and 2 in the main text rely solely on RMSE as the accuracy metric, which provides a limited evaluation of model performance.

2. In Table 3, some models lack performance results under the F1 metric.

3. The “Related Work” section merely lists existing studies without explicitly outlining the core challenges and innovations that motivated the design of the ICT method.

4. Tables 1-4 do not clearly highlight the optimal performance values.

5. Some abbreviations are used inconsistently or without proper definition—for example, “RMSE” is introduced without its full form in Table 1, and “ICTP” is used without definition at line 72 of the “Introduction” section.

---

> ### Author Rebuttal · Authors · 2026-03-28
>
> We thank the reviewer for the careful and constructive feedback. We appreciate your positive assessment of the motivation and potential of the ICT-based framework.
>
> **W1/Q1.** We agree that broader accuracy metrics would make the evaluation more complete. For the 3BPA system, which is particularly challenging in terms of extrapolation capability, most existing models (e.g., NequIP, MACE, Allegro, CACE, TACE, GRACE) report RMSE as the only standard evaluation metric. Using alternative metrics would make direct comparisons with these models difficult.  For the universal-model benchmark, however, we agree that the original table was too limited. In the revision, we will also provide a more complete set of reported metrics. The results are shown as follows.
>
> | Model | CPS↑ | Acc↑ | F1↑ | DAF↑ | Prec↑ | MAE↓ | R2↑ | κSRME↓ | RMSD↓ | Params↓ |
> |-------|------|------|-----|------|-------|------|-----|--------|-------|---------|
> |eSEN-30M-OAM|0.888|0.977|0.925|6.069|0.928|0.018|0.866|0.170|0.061|30.2M|
> |EquFlash | 0.888| 0.975 | 0.919 | 5.983 | 0.915 | 0.019 | 0.871 | 0.158 | 0.060 | 28.7M |
> |Nequip-OAM-XL| 0.886 | 0.971 | 0.906 | 5.869 | 0.897 | 0.020 | 0.872 | 0.125 | 0.063 | 32.1M |
> |MatRIS-10M-OAM| 0.877 | 0.976 | 0.921 | 6.039 | 0.923 | 0.019 | 0.871 | 0.218 | 0.060 | 10.4M |
> |Nequip-OAM-L| 0.870 | 0.967 | 0.893 | 5.823 | 0.890 | 0.022 | 0.865 | 0.166 | 0.065 | 9.6M |
> |**Cartesian OAM Model**| 0.867 | 0.965 | 0.889 | 5.749 | 0.879 | 0.022 | 0.865 | 0.173 | 0.065 | 18.8M |
> |GRACE-2L-OAM-L| 0.865 | 0.964 | 0.883 | 5.840 | 0.893 | 0.022 | 0.862 | 0.169 | 0.064 | 26.4M |
> |ORB v3| 0.860 | 0.971 | 0.905 | 5.912 | 0.904 | 0.024 | 0.821 | 0.210 | 0.075 | 25.5M |
> |Allegro-OAM-L| 0.840 | 0.966 | 0.895 | 5.674 | 0.867 | 0.022 | 0.868 | 0.319 | 0.065 | 9.7M |
> |GRACE-2L-OAM| 0.837 | 0.963 | 0.880 | 5.774 | 0.883 | 0.023 | 0.862 | 0.294 | 0.067 | 12.6M |
> |DPA-3.1-3M-FT| 0.802 | 0.963 | 0.884 | 5.667 | 0.866 | 0.023 | 0.869 | 0.470 | 0.069 | 3.27M |
> |eSEN-30M-MP| 0.797 | 0.946 | 0.831 | 5.260 | 0.804 | 0.033 | 0.822 | 0.340 | 0.075 | 30.1M |
> |MACE-MPA-0| 0.795 | 0.954 | 0.852 | 5.582 | 0.853 | 0.028 | 0.842 | 0.412 | 0.073 | 9.06M |
> |MatRIS-10M-MP| 0.778 | 0.951 | 0.847 | 5.422 | 0.829 | 0.031 | 0.824 | 0.489 | 0.072 | 10.4M |
> |AlphaNet-v1-OMA| 0.769 | 0.968 | 0.901 | 5.747 | 0.879 | 0.024 | 0.831 | 0.644 | 0.079 | 4.65M |
> |MatterSim v1 5M| 0.767 | 0.959 | 0.862 | 5.852 | 0.895 | 0.024 | 0.863 | 0.575 | 0.073 | 4.55M |
> |GRACE-1L-OAM| 0.761 | 0.944 | 0.824 | 5.255 | 0.803 | 0.031 | 0.842 | 0.517 | 0.072 | 3.45M |
> |Eqnorm MPtrj| 0.756 |0.929 | 0.786 | 4.844 | 0.741 | 0.040 | 0.799 | 0.408 | 0.084 | 1.31M |
> |Nequix MP PFT| 0.755 | 0.914 | 0.748 | 4.479 | 0.685 | 0.044 | 0.784 | 0.307 | 0.087 | 708k |
> |Nequip-MP-L| 0.733 | 0.921 | 0.761 | 4.704 | 0.719 | 0.043 | 0.791 | 0.452 | 0.086 | 9.6M |
> |Nequix MP| 0.729 | 0.914 | 0.751 | 4.455 | 0.681 | 0.044 | 0.782 | 0.446 | 0.085 | 708k |
> |Allegro-MP-L| 0.720 | 0.915 | 0.751 | 4.516 | 0.690 | 0.044 | 0.778 | 0.504 | 0.082 | 18.7M |
> |DPA-3.1-MPtrj| 0.718 | 0.936 | 0.803 | 5.024 | 0.768 | 0.037 | 0.812 | 0.650 | 0.080 | 4.81M |
> |SevenNet-l3i5| 0.714 | 0.920 | 0.760 | 4.629 | 0.708 | 0.044 | 0.776 | 0.550 | 0.085 | 1.17M |
> |HIENet| 0.707 | 0.929 | 0.777 | 4.932 | 0.754 | 0.041 | 0.793 | 0.642 | 0.080 | 7.51M |
> |GRACE-2L-MPtrj| 0.681 | 0.895 | 0.691 | 4.163 | 0.636 |0.052|0.741 | 0.526 | 0.090 | 15.3M |
> |MACE-MP-0| 0.637 | 0.878 | 0.669 | 3.777 | 0.577 | 0.057 |0.697 |0.682 | 0.092 | 4.69M |
> |eqV2 M| 0.558 | 0.975 | 0.917 | 6.047 | 0.924 | 0.020 | 0.848 |1.771 |0.069 | 86.6M |
> |ORB v2| 0.528 | 0.965 | 0.880 | 6.041 | 0.924 | 0.028 | 0.824 |1.734 |0.097 | 25.2M |
> |eqV2 S DeNS| 0.522 | 0.939 | 0.815 | 5.042 | 0.771 | 0.036 |0.788 |1.676 | 0.076 | 31.2M |
> |ORB v2 MPtrj| 0.470 | 0.922 | 0.765 | 4.702 | 0.719 | 0.045 |0.756 |1.726 | 0.101 | 25.2M |
> |CHGNet| 0.343 | 0.851 | 0.613 | 3.361 | 0.514 | 0.063 | 0.689 |2.000 |0.095 | 413k |
> |M3GNet| 0.310 | 0.812 | 0.569 | 2.882 | 0.441 | 0.075 | 0.585 | 2.000 |0.112 | 228k |
> |GNoME| NaN | 0.948 | 0.829 | 5.523 | 0.844 | 0.035 | 0.785 | n/a | n/a |16.2M|
>
> **W2/Q2.** The reason of missing F1 entries is that some of these models belong to the OMat24 series rather than the sAlex/MPtrj-based OAM series, so the corresponding stability classification is not directly comparable under the same labeling protocol. In contrast, the $\kappa$ metric is more robust and can be compared across different levels of computational fidelity.
>
> **W3/Q3.** We agree that the Related Work section does not state the motivating gap clearly enough. In the revision, we will reorganize the Introduction and Related Work sections to make the central challenge more explicit: ICT models lies in the fact that irreducible components are intrinsically mixed in Cartesian space. We will also revise the Related Work section to state the innovations that motivate our ICT framework.
>
> **W4/Q4/W5/Q5** We will address these issues in the revised version.

---

> > ### Author Rebuttal · Reviewer_GSPB · 2026-04-03
> >
> > Thank you for the authors' response. The additional metrics table in W1/Q1 and the explanation in W2/Q2 have resolved my concerns. The remaining issues have been promised to be addressed by the authors, and I acknowledge that. In summary, I will keep my current score.

---

> > > ### Author Response · Authors · 2026-04-03
> > >
> > > We are glad that that our rebuttal addressed your concerns. In the revision we will also improve the presentation more broadly by sharpening the motivating gap in the Related Work section, improving the table formatting and highlighting, and ensuring that abbreviations and notation are defined consistently throughout. We sincerely appreciate your constructive feedback, which has helped us strengthen both the evaluation and the presentation of the paper.

---

### Official Review · Reviewer_ZrNQ · 2026-03-12

**Soundness:** 2
**Presentation:** 3
**Significance:** 2
**Originality:** 2
**Overall Recommendation:** 3
**Confidence:** 3

**Summary:**

This paper formalizes the use of irreducible Cartesian tensor representations for O(3)‑equivariant architectures by introducing Cartesian‑3j and ‑nj symbols that provide Cartesian counterparts of the spherical harmonic Clebsch–Gordan coefficients required for tensor product computation. The authors use this to extend the e3nn library and implement Cartesian counterparts of NequIP, Allegro, and MACE. Based on the OMat24/sAlex/MPtrj datasets, the Cartesian architectures achieve comparable accuracy but show no improvement in memory usage or compute time.

**Compliance With Llm Reviewing Policy:**

Affirmed.

**Key Questions For Authors:**

- The F1 metric (stable/unstable material classification) should be defined in text
- Table 1: It is unclear if the reported results are for Allegro or NequIP.
- Section 5: The authors mention the library oeq, but do not provide any source or link to, e.g., the home page.
- Section 4.3, what is the U tensor? Please define and give an example
- Section 4.4, what is k? Please define.
- Section 4.5, what is i? Should it be k?

**Limitations:**

yes

**Strengths And Weaknesses:**

Strengths
- A formalization and general implementation of Cartesian tensor operations could accelerate research on Cartesian MLPs. The results presented suggest that Cartesian models can potentially achieve the same accuracy.
- The authors fairly discuss the limited computational efficiency of Cartesian tensor representations (their work) for higher-order tensors and as a replacement for spherical representations in existing architecture.

Weaknesses
- Besides its implementation and verification in experiment 3, the paper is only incremental over previous work. Much of the theory, except for the introduced 3j and nj symbols, is already given in, e.g., Xu et. al. (2025) arxiv.org/pdf/2509.14961. The authors also adopt section 3 (background) nearly unchanged from this work.
- Comparison with previously published universal models is limited to just two metrics F1 and Thermal Conductivity Metrics. Accuracy with respect to other properties (e.g., all metrics reported by Matbench Discovery, energy, and force, etc.) is necessary to support the author’s claim of comparable accuracy of Cartesian models compared to existing ones.
- The presented Cartesian models do not offer an advantage over existing models that predominantly rely on spherical-harmonic representations. Models with the same architecture but a different basis reach the same accuracy on the reported metrics, but with a small to pronounced increase in memory requirements and runtime.
- The authors test the correctness of the implementation by replacing spherical tensor representations with Cartesian representations in three different models and demonstrate that the models converge to approximately the same accuracy. A more strategic approach to test the correctness would be to compare the results of tensor products or contractions for a series of (random) test tensors.
- "By contrast, applying analogous strategies in spherical models, that is, restricting STP paths to mimic the output of Cartesian models, results in lower accuracy, underscoring a unique advantage of Cartesian representations." No explanation, experiments, or references are given for that claim.

---

> ### Author Rebuttal · Authors · 2026-03-27
>
> We thank the reviewer for the constructive comments.
>
> **W1.** We agree that some background theory in the manuscript is already discussed in Xu et al. (2025), and we will revise the paper to attribute and delimit that material more clearly. However, we do not think the present submission is simply incremental over that work. Xu et al. (2025) is primarily an Cartesian ACE architecture paper with universal embeddings and broad application settings. By contrast, the main contribution here is to introduce the missing coupling objects needed to make ICT usable in mainstream equivariant architectures: Cartesian-3j, generalized Clebsch-Gordan coefficients in Cartesian space (nj), and an e3nn-style implementation. These, in turn, enable Cartesian counterparts of NequIP, Allegro, and MACE, and the first controlled basis-level comparison in which the network architecture is held fixed and only the tensor basis is changed. This is important because prior comparisons between Cartesian and spherical models were typically confounded by simultaneous changes in architecture, operator design, and implementation details, making it difficult to disentangle basis effects from architectural and engineering effects. By isolating this factor, the present work provides a more principled way to understand the actual role of Cartesian representations and to guide future Cartesian and hybrid Cartesian-spherical model design.
>
> **W2.** We have added remaining metrics from Matbech, please refer them at **W1/Q1** by *reviewer 4*. In particular, the table now reports results on the Unique Prototypes subset, which provides a more objective evaluation compared to the Full Test Set and 10k Most Stable subsets in the MatBenchbenchmark. Regarding energy and force errors, a direct comparison is not available. The OAM model is trained on sAlex/MPtrj datasets, but there is no universally agreed-upon validation and test split across all methods. Although the official sAlex dataset provides a predefined validation split, it is difficult to ensure that all competing models follow the same data partitioning protocol, which limits the fairness of direct comparisons.
>
> **W3.** This is consistent with the main message of our experiments: naive basis substitution is not sufficient and this is why we further explore the design space of Cartesian models in this work.
>
> **W4.** This validation was indeed performed during development using ICT decomposition. In the revision, we will make this numerical verification clearer and provide the corresponding scripts in the supplementary material.
>
> **W5.** We agree that the original sentence might be too strong as written. Under the same architectural setting, our dedicated Cartesian model is competitive with the corresponding spherical model and can be faster. Since no existing model natively operates in both representations within a single implementation, we use our matched Cartesian and spherical variants for comparison. The evaluation metrics below use the same units as in Tables 1 and 2 of the main text.
>
> | 3BPA-Metric|cartModel|sphModel|
> |---|---|---|
> |E-300K↓|2.89|2.90
> |F-300K↓|8.68|9.07
> |E-600K↓|8.10|10.83
> |F-600K↓|19.88|22.02
> |E-1200K↓|23.42|28.03
> |F-1200K↓|54.26|61.04
> |E-dih↓|11.0|7.26
> |F-dih↓|16.45|16.62
> |Speed↓|9.7|16.4
> |Memory↓|5490|3640
> |Params ↓|4.1M|4.1M
>
> This comparison supports the broader architectural point more directly. We will revise that sentence to avoid overstating the evidence.
>
> **Q1.** The F1 score is defined as the harmonic mean of precision and recall:
>
> $$
> F_1 = 2 \cdot \frac{\mathrm{Precision} \cdot \mathrm{Recall}}{\mathrm{Precision} + \mathrm{Recall}},
> $$
>
> where precision and recall are computed based on the binary classification of stable versus unstable structures. Specifically, the classification of stable and unstable structures is based on the energy above the convex hull.
>
> **Q2.** We will revise the table caption and column labels, the last two columns of Table 2 correspond to Allegro, while all other columns correspond to NequIP.
>
> **Q3.** oeq has already been cited in the main text; please refer to 475–479 col 1.
>
> **Q4.** We will include a more detailed explanation in the supplementary materials. Here, we provide a brief description.
> In practice, the U matrix corresponds to the generalized Clebsch-Gordan coefficients themselves, referred to in this work as Cartesian-nj. Due to the combination of different coupling paths, each generalized CG coefficient has a different shape. To enable vectorized computation, multiple paths were stacked together and apply zero padding. As a result, the U matrix has shape $(3^{L_{out}}, (\sum_l^{L_{in}} 3^l,)^{correlation},num path)$, which corresponds to the `U_tensor_real` implementation in the MACE code.
>
> **Q5/6.** Here, $k$ denotes the number of contraction axes. We believe the reviewer is referring to the definition of $i$ in the supplementary materials, where it has already been specified that $i$ corresponds to the target atom index.

---

> > ### Author Rebuttal · Reviewer_ZrNQ · 2026-04-01
> >
> > My questions have been addressed.

---

> > > ### Author Response · Authors · 2026-04-02
> > >
> > > We thank the reviewer for confirming that our rebuttal addressed your concerns. Your comments helped us see more clearly where the paper needed to be sharper.
> > >
> > > Most importantly, we will refine the paper’s scope so that it is framed as introducing the Cartesian coupling primitives that enable a controlled Cartesian-vs.-spherical comparison under matched architectures. We will also clarify the relation to prior Cartesian work and make the paper’s novelty and motivating gap more explicit. On the empirical side, we will broaden the benchmark reporting and make the numerical verification clearer, including the corresponding supplementary scripts.
> > > We will also revise the paper to make sure it now consistently emphasizes what it does establish: a principled controlled comparison, the viability of the Cartesian framework, and the need for dedicated Cartesian design choices.
> > >
> > > Beyond these main revisions, we will thoroughly revise the paper for readability. Thank you again for helping us strengthen the paper, and if you feel these changes improve the submission, we would be grateful if you would take that into account in your final assessment.

---

### Official Review · Reviewer_oseT · 2026-03-12

**Soundness:** 4
**Presentation:** 4
**Significance:** 3
**Originality:** 4
**Overall Recommendation:** 5
**Confidence:** 3

**Summary:**

The paper explores a new approach for constructing equivariant neural network architectures, particularly for application to machine learning interatomic potentials. Whereas most equivariant models utilize spherical tensors the authors present a new framework for efficient construction of equivariant neural network layers based on Cartesian tensors.

The approach is builds heavily upon a recent work describing efficient constructions of irreducible Cartesian tensor (ICT) decompositions of arbitrary rank (Shao et al., 2025), which introduces analytical matrix representations of 'irreducible Cartesian tensor decomposition' (ICTD) operators. The matrix representation of the ICTD operator acting on a 'vectorized' / 'flattened' representation of a general Cartesian tensor yields a similarly vectorized irreducible component of the tensor.

The paper introduces Cartesian 3j symbols in analogy to the Wigner 3j symbols via the 'path matrix' construction introduced by Shao et al. These act as the coefficients of a linear expansion of an reducible Cartesian tensor product in terms of its irreducible components. The path matrix construction suggests an intuitive explanation that the ICTD operator acts by 'round-tripping' via the spherical basis -- a reducible tensor is projected into the spherical basis, the components are combined under the usual selection rules, and projected back into the Cartesian basis. Thus an irreducible Cartesian tensor product (ICTP) can be constructed by taking the reducible Cartesian tensor product (RCTP) and applying the ICTD operator.

The MACE architecture introduced an efficient scheme for implementing tensor contractions over features in the spherical basis using generalized Clebsch-Gordan coefficients. Here, the authors demonstrate that a similarly efficient contraction can be implemented by using the introduced Cartesian-3j symbols to obtain generalized Clebsch-Gordan coefficients in the Cartesian basis (referred to as Cartesian-nj).  At this point, the authors have now demonstrated that under their scheme, they can carry out efficient tensor products / contractions in the cartesian basis. They note that introducing non-linear layers can be handled identically in the Cartesian and spherical cases (and present the particular scheme they employ). With these tools in hand, they are able to implement equivariant neural network architectures.

The remainder of the paper is dedicated to numerical experiments comparing spherical and Cartesian equivariant network implementations. Initially, the authors compare the performance of spherical equivariant architectures converted into directly analogous Cartesian ones. They find that this results in identical (up to a reasonable margin to be considered as implementation subtleties) results. These results suffice to demonstrate the validity of the framework, but demonstrate that the Cartesian representation naively 'swapped-in' for the spherical representation does not yield a benefit in terms of memory or compute efficiency.

Next, the authors point out several design considerations that should be applied in constructing *new* architectures using the Cartesian basis. They leverage the fact that the RCTP (or contraction, RCTC) can be used when applying edge operations, while retaining the tensor structure. Tensor products between node features are computed on edges, passed through a linear layer, and aggregated to nodes; then, the node features are projected back into the irreducible components via the ICTD operators.

These ideas are used to construct a new model architecture. The authors detail several numerical experiments with this architecture (simply referred to as 'Cartesian model') on various open datasets. Their models are shown to yield performance which is competitive with (although, notably, not exceeding) state of the art equivariant architectures. The performance of the architecture is discussed, noting that an equivalent architecture in the spherical basis would be less performant due to the requirement of applying the Wigner-3j symbols in the edge product / contractions (which can be skipped in the Cartesian case, favouring later projection at the node level). The authors note that for equivariant models in the spherical basis, there are tailored operator-fusion libraries which can be used to reduce memory and compute footprints which do not yet exist for the Cartesian basis.

**Compliance With Llm Reviewing Policy:**

Affirmed.

**Final Justification:**

My evaluation of the paper has not changed as a result of the rebuttal: I had only minor concerns which have been addressed.

**Key Questions For Authors:**

1) How does the performance (both in terms of computational cost and resulting accuracy) of the developed Cartesian models compare to existing Cartesian models (e.g. TACE)?
2) Are there any obvious ways in which, with the advantages of new approach in mind, operating in the Cartesian basis produces models that have a clear advantage over the spherical basis?

**Limitations:**

yes

**Strengths And Weaknesses:**

The quality of the submission is very high: the authors set out with a clearly defined goal (introduce an analogy of the Wigner-3j and nj symbols in the Cartesian basis). The relevant theory is clearly laid out, and the solution (based on a prior work) presented in a straightforward manner. The benefits of this solution are described, and compared throughout to the analogous concepts in the spherical basis which will be more familiar to most readers. The relative strengths and weaknesses of the Cartesian and spherical representations are clearly explained and demonstrated through numerical experiments. Furthermore, guidelines for the practical application of the theoretical developments made by the paper are discussed.

### Soundness

The submission is technically sound. The introduced theory is closely derived from prior published results. Claims about model performance are modest, critical, and fully supported by the presented empirical results.

I have a single criticism: the authors (as mentioned in the submission) are not the first to introduce a model architecture capable of working in a Cartesian basis using irreducible components of arbitrary rank. However, all of the numerical experiments only compare the newly introduced Cartesian architecture to equivariant architectures using spherical bases. I think it would make the paper even stronger were the authors to compare the performance (both accuracy and computational cost) of their approach to previous approaches; particularly as they claim in the text that their approach should yield benefits in computational efficiency (due to the possibility of pre-computing the contraction symbols).

### Presentation

The submission is well presented. The narrative is easy to follow, the broad context of equivariant models is clearly discussed, all of the empirical data is laid out in clear tables.

I have a single nit pick: the symbol $k$ is introduced to represent the number of contracted indices (I think?) under the 'Empirical Observations' header in subsection 4.4. This symbol is not previously defined in the paper, and may be confusing to readers who are not entirely familiar with the topic (which would be a shame in an otherwise highly accessible paper).

### Significance

I believe the paper is of 'good' significance. The work opens up several lines of -- likely fruitful -- investigation into the architecture and design of models in the Cartesian basis. I refrain from claiming 'excellence' for the following reasons: in real terms, the developments introduced beyond the original theoretical contributions of Shao et al. are modest; additionally, the numerical experiments do not make it clear if, even within this new framework, equivariant models in the Cartesian basis will find practical application / advantages beyond existing equivariant models in the spherical basis.

### Originality

This paper is highly original. It introduces a (to my knowledge) entirely novel approach for constructing computationally efficient tensor products and contractions in the space of irreducible Cartesian tensors to the field of equivariant neural networks.

---

> ### Author Rebuttal · Authors · 2026-03-27
>
> We thank the reviewer for the constructive and encouraging feedback. We have carefully considered your comments and will incorporate the necessary revisions in the revised version.
>
> **W1/Q1.** We acknowledge that the current manuscript does not include a direct comparison across different Cartesian network architectures, and we agree that such a comparison would further strengthen the paper. The main reason is that the present work is primarily focused on comparing Cartesian and spherical formulations under closely matched architectural settings, rather than benchmarking all existing Cartesian models against one another. In addition, Cartesian models supporting irreducible representations of arbitrary rank are still relatively limited. Existing related approaches differ substantially not only in implementation, but also in their underlying architectural assumptions. For example, ICTP is theoretically formulated for arbitrary rank, but its publicly available implementation is limited to rank-3 as it is based on a tensor product framework dating back to 1989, where each contraction should be writen manually. CartNet, to our knowledge at the time of writing, is available only as an arXiv preprint and does not yet provide an official open-source implementation. These practical limitations already make broad empirical comparison nontrivial.
>
> A further difficulty is that the compared Cartesian approaches are not built on the same tensor-product strategy. Following MACE, our method adopts a precomputed product-basis formulation, which can provide a substantial speed advantage over approaches such as TACE in low-correlation settings, but at the cost of increased memory usage when higher-order correlations (≥3) are considered. In particular, for correlation=2 with multiple stacked layers, the precomputed basis can significantly accelerate the computation of uncoupled product bases in both Cartesian and spherical settings. However, this strategy is not well suited to coupled product bases, because the intermediate tensors become prohibitively large. By contrast, TACE and ICTP rely more heavily on coupled product-basis constructions, whereas Cartesian-nj follows the same uncoupled product-basis philosophy as MACE. At the same time, our models adopt nonlinear interaction block (where nonlinearity is crucial when training uMLIP in our paper), wheras TACE not adopt in therir paper. This difference may make our architecture more favorable in large-scale datasets that require stronger nonlinear expressivity. As a result, a fully fair comparison under identical architectural assumptions is not straightforward. We will revise the manuscript to explain these differences more clearly and to better position our method relative to prior Cartesian approaches in terms of both architectural design and practical trade-offs.
>
> **Q2.** Based on our current implementation, under the same architecture without specialized operator fusion libraries, Cartesian models are faster than spherical models when $L \leq 3$. This is also a practically important regime, since
> $L=3$ is already widely used in high-performing equivariant models on mainstream benchmarks such as 3BPA, rMD17, MD22, QM9, and liquid water. The main reason is that, in our architecture, Cartesian tensors admit the coupling cost to be shifted away from edge-level operations. In this sense, the clearest current advantage of the Cartesian basis is not universal superiority, but a more favorable organization of tensor operations in low-rank settings.
>
> On the other hand, atomic coordinates and many tensorial targets are naturally expressed in Cartesian form, which also makes the representation conceptually appealing. We expect that improved Cartesian operator-fusion libraries would further strengthen the low-rank advantage observed here, although they would not remove the fundamental high-rank growth of Cartesian tensors.

---

> > ### Author Rebuttal · Reviewer_oseT · 2026-04-02
> >
> > The highlighted complexities in producing apples-to-apples comparisons with other implementations of equivariant architectures in Cartesian bases are substantial, and the authors commitment to clarifying these in the text satisfies my original question.
> >
> > Likewise, the response to my second question is satisfactory and I think readers who are more familiar with the topic will already understand this from the original text.

---

> > > ### Author Response · Authors · 2026-04-02
> > >
> > > We thank the reviewer for the careful reading, encouraging assessment, and for confirming that our rebuttal addressed your questions.
> > >
> > > We will better position the paper relative to prior Cartesian approaches by making the architectural differences and tensor-product trade-offs more explicit, and by clarifying why direct apples-to-apples comparison across existing Cartesian implementations is nontrivial. We will also sharpen the discussion of where the current Cartesian framework does and does not offer an advantage. We will also carefully revise the manuscript for notation and readability throughout.
> > >
> > > Thank you again for the supportive and constructive feedback.

---

### Official Review · Reviewer_Rifm · 2026-03-13

**Soundness:** 3
**Presentation:** 3
**Significance:** 2
**Originality:** 3
**Overall Recommendation:** 4
**Confidence:** 4

**Summary:**

This paper explores the feasibility of using irreducible Cartesian tensors (ICTs) instead of spherical tensors (STs) to construct O(3) equivariant neural networks. They define Cartesian-3j, an analog of the Wigner-3j symbol and Cartesian-nj as an analog of generalized Clebsch-Gordan coefficients from MACE. This enables a completely Cartesian implementation of standard equivariant models NequIP, MACE, and Allegro. In addition, Cartesian tensors enable a contraction operation which can keep tensor ranks small and has no obvious analog in the spherical case. Empirically, they demonstrate simply converting a spherical tensor based model into a Cartesian one offers no benefits. Instead, one must design a separate architecture leveraging the properties of Cartesian tensors. They show such models are competitive with existing ones though they do not seem to stand out.

**Compliance With Llm Reviewing Policy:**

Affirmed.

**Final Justification:**

The authors' responses addressed most of my concerns and clarified the intended framing, it is not to provide some new SOTA method but to enable fairer comparisons between spherical and Cartesian frameworks. Ultimately, the results are not too surprising, a model designed for spherical tensors performs worse when replaced with Cartesian tensors and vice versa. However, I do think they are important and informative so I raise my score from 3 to 4.

**Key Questions For Authors:**

* The Cartesian-3j is defined using the first multiplicity of irrep $\ell_3$. Does this introduce normalization issues? In particular, it seems like it may be possible to trace the corresponding path and “undo” the normalization issue so it agrees exactly with Wigner-3j. Is this considered?
* How much improvement would you expect from specialized libraries to accelerate Cartesian tensors (similar to cueq, equflash, etc. for spherical tensors)?
* ICTC has some resemblance to FusedTensor in e3x. I would be curious if the authors had any thoughts on their relationship.
* The Wigner-nj symbols for higher n refer to a change of basis between different coupling orders in physics context, not the generalized Clebsch-Gordan of MACE. Perhaps consider renaming Cartesian-nj to avoid confusion.
* 181 - I believe these empirical observations are directly provable. No matter which $k$ is chosen, ICTC is still a bilinear operation. Therefore, by the universality property of (full) tensor products ICTC can always be expressed as the composition of a tensor product and an equivariant linear layer. Since weight $\ell_3$ appears only once in the Clebsch-Gordan tensor product, by Schur’s lemma any $\ell_3$ irrep in the output must be a multiple of that one $\ell_3$ irrep in the tensor product. This proves observations 1 and 2. For 3, the final tensor is rank $\ell_1+\ell_2-2k$ giving the upper bound. The lower bound follows from the fact that the triangle inequality must be satisfied in Clebsch-Gordan tensor products.

**Limitations:**

yes

**Strengths And Weaknesses:**

## Strengths
* The paper is well organized and clearly describes the different components that can be used for a Cartesian tensor based network
* The question of what framework is best for equivariant networks and their individual tradeoffs is an important one.
* The Cartesian 3j and nj constructions are novel and allow the first direct comparison of spherical and Cartesian tensor based networks
* The experiments are illustrative and agree with the theory that maps spherical methods to Cartesian ones

## Weaknesses
* The main weakness is that empirically Cartesian based networks do not seem to offer any significant benefits over the standard spherical ones. This is not too surprising if we directly map an architecture designed for spherical tensors onto Cartesian tensors. However, even the specially designed Cartesian network does not seem to perform better than existing spherical ones.
* It would be nice to see if a hybrid architecture using both spherical and Cartesian tensors (in particular leveraging ICTC) could perform better.
* I would like to see more discussion of the normalization of the Cartesian-3j and especially how such choices would affect the Cartesian-nj
* It seems the main potential benefit of Cartesian tensors is the contraction operation. However, it is not clear to me how helpful this is from the current experiments.

## Minor issues
* 162 col 2 - STP undefined (I assume Spherical Tensor Product)
* 165 col 1 - “contracion”
* 215 col 1 - ICTC undefined (I assume Irreducible Cartesian Tensor Contraction)
* 186 col 2 - $k$ undefined (I assume number of indices contracted over)
* 223 col 2 - I do not see the speed benefit in the tables
* 238 col 2 - Seems a bit misleading, they can be combined through linear transformations but it is inconvenient when they are embedded in Cartesian tensors of different rank.
* 257 col - “contracion”
* Making bold the better performing choice in the tables would make them easier to read
* Add units to the table not just description, for instance Speed (s) instead of just Speed, also arrow for whether higher or lower is better

---

> ### Author Rebuttal · Authors · 2026-03-27
>
> We appreciate the recognition of the novelty of the Cartesian framework and insightful evaluation of our work.
>
> **W1.** We agree that the results do not support a blanket claim that Cartesian models already outperform spherical ones. This is not our intended claim. Rather, the key contribution is to establish a general Cartesian-3j framework, to enable the first controlled basis-level comparison under matched architecture. In that sense, the negative result of direct “Cartesianization” is itself informative, and dedicated Cartesian architectural choices are needed. Also, the uMLIP comparison is carried out under constrained settings (48 channels, 5 layers, $(L=3)$, restricted tensor-product scheme $(l_1 \leq l_2)$). We therefore view the current results as evidence about the viability of the Cartesian formulation, rather than as a definitive statement about its ceiling. We will revise the manuscript to make this scope clearer and to avoid any wording that could be read as claiming general empirical superiority over spherical models.
>
> **W2.** We agree that hybrid architectures are promising. We will highlight this in the outlook.
>
> **W3/Q1.** After transformation to ST, these alternatives become linearly dependent and differ only by a constant factor. We therefore follow the parentage ordering and select the first occurrence of $l_3$ . Different scalings can affect training stability. For this reason, we normalize Cartesian-3j to unit $L_2$ norm, consistent with the convention used for Wigner-3j in e3nn. For Cartesian-nj, once the Cartesian-3j normalization is fixed, we adopt the same treatment as in MACE; in ICTC, each path output is further scaled by $1/\sqrt{3^k}$. More broadly, we agree that normalization in Cartesian space deserves further study. Unlike nonzero ST, Cartesian tensors are subject to symmetric-traceless constraints, so some components are structurally zero, making scale less transparent. A more principled normalization may therefore exist, but the present $3^l$-based treatment has been effective in practice.
>
> **W4/Q2.** In our architecture, the value of ICTC is not only a possible accuracy gain, but also the different way it organizes tensor operations. ICTC can be combined with ICTP at the edge level and irreducible projection can be deferred to the node level. This helps reduce edge-level coupling overhead, and preserves information that would otherwise be discarded in a highest-weight-only scheme. Empirically, adding ICTC on top of ICTP improves the accuracy of our highest-weight model. More generally, grid-based tensor-product methods in the spatial domain also use restricted subsets of tensor-product paths and omit antisymmetric interactions such as the cross product, yet such sparsified schemes can still be effective in practice. We therefore believe that the principles governing sparse tensor-product path selection remain insufficiently understood and deserve further study.
>
> This also helps explain why the current tables should not be interpreted as the ceiling of Cartesian methods. In particular, our architecture is designed so that the main advantage of the Cartesian formulation appears in where the coupling is performed, rather than as a uniform speedup. Prior works (HotPP, ICTP) have shown that Cartesian tensor operations can be advantageous in moderate angular regimes, roughly $(L \leq 4)$, which is consistent with our results. Also, improved libraries would likely narrow the current implementation gap, although they would not remove the fundamental high-rank growth of Cartesian tensors.  For the accuracy and speed comparison between our matched Cartesian and spherical models, please refer to **W5** by *Reviewer 3*.
>
> **Q3.** We carefully examined the FusedTensor in e3x. After factoring out the batch/channel dimensions and learnable parameters, the operation can be understood as `ST -> matrix -> matrix @ matrix -> ST`. This operation can be fused into a single einsum expression`(c, abc, e, bde, adf -> f, in1, cg1, in2, cg2, cg3)`. This shows the operation is equivalent to multiplying the two input tensors by a constant coefficient (It's actually the CG coefficient itself) and thus FusedTensor is essentially an alternative implementation of CGTP (maybe faster in some L range), rather than an ICTC operation.
>
> **Q4.** We will adopt the more precise term “generalized Clebsch-Gordan coefficients in Cartesian space”
>
> **Q5.** We have carefully reviewed the proof and agree that the reasoning appears sound. However, we do not yet feel sufficiently confident in our own understanding to present these statements as formal theoretical results in the current manuscript. To avoid overstating the case or potentially misleading readers, we prefer to keep them as empirical observations for now, and leave a more complete proof to future work.
>
> We will address the minor issues accordingly.

---

> > ### Author Rebuttal · Reviewer_Rifm · 2026-04-02
> >
> > I thank the authors for their detailed rebuttal which has clarified many points. I definitely think the framing can be improved by emphasizing the necessity of fairly comparing Cartesian and spherical frameworks, and that the Cartesian-3j and nj enable this. Then it becomes clear that even a negative result is interesting. Currently the paper reads as a standard “we introduce a new method and test it” work where one expects the new method to give substantial improvements.
> >
> > I have a few followup questions mostly to check my understanding.
> >
> > **W3** In this case, is it correct that equation (6) gets rescaled in practice? If so, I think it should be mentioned in the main text and maybe even written explicitly in the equation. Also, is this $L_2$ norm done separately for each $\ell_3$?
> >
> > **W4/Q2** Yes, this makes sense. For the table in **W5** by Reviewer 3, it is essentially a model designed for Cartesian components which performs worse when you swap in spherical components correct?
> >
> > **Q3** The reason I ask is because `FusedTensor` and ICTC seem to have a “path merging” property (different paths leading to the same output irrep type get “averaged” together). I also believe `FusedTensor` up to $\ell=2$ gives a 3x3 matrix, which corresponds to a rank 2 Cartesian tensor. Hence the matrix multiplication should correspond to an ICTC in this case. In addition, the spatial grid based methods (which I believe is gaunt tensor product [1]) also has this “path merging” property. I believe some of these other “path merging”methods were analyzed in [2] and I think it could be helpful to have a discussion in the appendix.
> >
> > # References
> > [1] Luo, S., Chen, T., & Krishnapriyan, A. S. Enabling Efficient Equivariant Operations in the Fourier Basis via Gaunt Tensor Products. In The Twelfth International Conference on Learning Representations.
> >
> > [2] Xie, Y., Daigavane, A., Kotak, M., & Smidt, T. The Price of Freedom: Exploring Expressivity and Runtime Tradeoffs in Equivariant Tensor Products. In Forty-second International Conference on Machine Learning.

---

> > > ### Author Response · Authors · 2026-04-03
> > >
> > > We thank the reviewer for the helpful follow-up. We will carefully revise the paper to sharpen its fouces on introducing Cartesian-3j and Cartesian generalized Clebsch-Gordan coefficients that enable a fair Cartesian-vs.-spherical comparison under matched architectures.
> > >
> > > We are also delighted to take this opportunity to discuss more regarding the comments.
> > >
> > > **W3.** Yes. Equation (6) is indeed normalized in practice by dividing it by its L2 norm, and we agree this should be stated more explicitly in the main text. For fixed $\ell_1$ and $\ell_2$ , the same output $\ell_3$ can appear with multiple multiplicities. After transformation to spherical space, these copies are linearly dependent and differ only by a scalar factor. We therefore follow the parentage ordering, choose the first occurrence as the canonical representative, and normalize that representative to unit L2 norm.
> > >
> > > **W4/Q2.** Yes, the model was intially designed for Cartesian components. For the table, the architecture and training setup are held fixed, and the representation/coupling is swapped. Under that matched comparison, the Cartesian-oriented version gives the stronger overall trade-off on most reported 3BPA metrics. We do observe that using the highest-weight ICTP and ICTC leads to some improvement in accuracy. In terms of model architecture, the basic components remain linear layers and nonlinearity (MLP and 0e/Norm Gate), so we believe the improvement in accuracy can be attributed to the Cartesian operation itself.
> > >
> > > **Q3.** We agree there is a meaningful connection here through path merging. However, we believe that the usage of tensor products and contractions in the Cartesian framework deserves further clarification. In our current Cartesian design, we keep a single highest-weight component for each operation, the optimal choice is to employ the highest-weight components in both ICTP and ICTC. Specifically, given two input tensors of rank $l_1$ and $l_2$, and contracting over $k$ indices:
> > >
> > > - For $k = 0$ (ICTP), we retain only the component with weight $l_1 + l_2$.
> > > - For $k > 0$ (ICTC), we retain only the component with weight $l_1 + l_2 - 2k$.
> > >
> > > This selection ensures that the number of model parameters in the Cartesian space exactly with spherical space in SO(3). From that viewpoint, CGTP are the most unconstrained construction, since they can assign independent learnable weights to each admissible path. As also discussed in the implementation of FusedTensor, path merging  is like assigning identical weights to paths that share the same $l_3$ output. Under this constraint, both the FusedTensor approach and the Gaunt tensor product achieve excellent accuracy.
> > >
> > > We also agree that the relevant references should be discussed more explicitly. We previously referred to these approaches as spatial grid-based methods, based on our own experimental observations that they lead to a notably simple computational scheme. However, we later became aware that related ideas had already been explored in ref [1], which you mentioned. Nevertheless, in general, the Gaunt tensor product can also be implemented in the Cartesian framework, and we have already verified this through our experiments. In a recent work “Asymptotically Fast Clebsch-Gordan Tensor Products with Vector Spherical Harmonics", they address antisymmetry using tensor signals defined on grids. We also emphasize that in our model ICTP and ICTC have independent learnable weights, rather than same parameter sharing across different path.
> > >
> > > We will therefore add an appendix discussion regarding these.
> > >
> > > Thank you again for these suggestions. They will improve both the framing and the technical clarity of the revised manuscript.

---

### Decision · Program_Chairs · 2026-04-30

**Decision:**

Accept (regular)

**Comment:**

The authors present a way to construct equivariant neural networks using irreducible Cartesian tensors, replacing the irreducible spherical tensors used in standard MLIPs like MACE, Nequip and Allegro. This is implemented using a construction recently introduced in Shao et al 2025. The authors show that when plugged directly into existing MLIP architectures, the novel construction does not lead to any improvement. They then show that the novel construction can inspire new architectures which may work better than existing architectures designed for spherical tensors. The reviewers raised concerns about the lack of improvement in empirical performance. All reviewers found the paper to be rigorous in its evaluation and honest in its conclusions. This is the rare paper that introduces a novel tool that can be used for *many* models, rather than simply introducing a new model and showing it improving on benchmarks (though the novelty here is not in the tool itself, but the application of the tool to MLIPs). That means the potential from this paper is in enabling many different novel models in the future, rather than introducing a single model that is potentially useful. For this reason, I am less concerned by the weak empirical results than I normally would be. Although strong empirical results would make the paper much better, I still recommend acceptance, given the future potential for what could be accomplished with these techniques.